# Angioplasty induces epigenomic remodeling in injured arteries

Mengxue Zhang[1],*, Go Urabe[1],*, Hatice Gulcin Ozer[2], Xiujie Xie[1], Amy Webb[2], Takuro Shirasu[1], Jing Li[1], Renzhi Han[3], K Craig Kent[1], Bowen Wang[1] , Lian-Wang Guo[1,4,5]

**Neointimal hyperplasia/proliferation (IH) is the primary etiology of vascular stenosis. Epigenomic studies concerning IH have been largely confined to in vitro models, and IH-underlying epigenetic mechanisms remain poorly understood. This study integrates information from in vivo epigenomic mapping, conditional knockout, gene transfer and pharmacology in rodent models of IH. The data from injured (IH-prone) rat arteries revealed a surge of genome-wide occupancy by histone-3 lysine-27 trimethylation (H3K27me3), a gene-repression mark. This was unexpected in the traditional view of prevailing post-injury gene activation rather than repression. Further analysis illustrated a shift of H3K27me3 enrichment to anti-proliferative genes, from pro-proliferative genes where gene-activation mark H3K27ac(acetylation) accumulated instead. H3K27ac and its reader BRD4 (bromodomain protein) co-enriched at *Ezh2*; conditional BRD4 knockout in injured mouse arteries reduced H3K27me3 and its writer EZH2, which positively regulated another pro-IH chromatin modulator UHRF1. Thus, results uncover injury-induced loci-specific H3K27me3 redistribution in the epigenomic landscape entailing BRD4→EZH2→UHRF1 hierarchical regulations. Given that these players are pharmaceutical targets, further research may help improve treatments of IH.**

## Introduction

Neointimal hyperplasia (IH) in the inner vascular wall obstructs blood flow engendering cardiovascular diseases. IH not only occurs in atherosclerosis, but persists after recanalization of stenosed arteries, causing re-stenosis. Drug-eluting stents and balloons are commonly deployed to impede post-angioplasty IH. However, they are unable to eradicate IH yet potentially thrombogenic, as exposed by multicenter meta-analyses (Holy et al, 2014; Byrne et al, 2015). The concerns culminated with three consecutive FDA warnings of increased mortality potentially associated with paclitaxel-eluting stents and balloons. A compelling agenda thus emerges to better understand IH pathogenesis for therapeutic improvement (Wang et al, 2018).

Neointima is primarily formed by vascular smooth muscle cells (SMCs) that have transitioned to a migro-proliferative state (Yoshida et al, 2008). With the same genome, without DNA sequence changes – this SMC state transition is epigenetic in its nature (Gomez et al, 2015; Wang et al, 2015; Das et al, 2017). Indeed, a number of epigenetic factors, mainly DNA and histone modification enzymes, are known players in SMC proliferation and IH (Gomez et al, 2015), and mostly studied with a focus on pro-proliferative gene activation (Marx et al, 2011; Byrne et al, 2015). Recently, BRD4, a bromo and extraterminal (BET) family histone mark reader and gene coactivator, was suggested to be a determinant of SMC proliferation and IH (Wang et al, 2015; Dutzmann et al, 2021) whereas this remained to be verified through tissue-specific BRD4 KO. BRD4's bromodomains read/bind acetylated histone sites (e.g., H3K27ac), with its C-terminal domain interacting with the transcription elongation complex (Borck et al, 2020). As such, BRD4 acts as a linchpin that couples cis- (e.g., enhancers) and trans-regulators to the central transcription machinery, thereby localizing this assembly to specific genomic loci to activate a select set of genes (Borck et al, 2020; Shi & Vakoc, 2014). Inasmuch as BETs/BRD4 inhibition stymies SMC proliferation but not endothelial growth, as evidenced both in vitro and in vivo (Brown et al, 2014; Wang et al, 2015, 2018; Zhang et al, 2019), BRD4 appears to be an attractive anti-restenotic target (Ostriker & Martin, 2015; Wang et al, 2018; Borck et al, 2020). However, whereas gene activation has been the mainstay of mechanistic studies on SMC/ neointima proliferation, gene repression is often overlooked.

As opposed to BRD4's role in transcriptional activation (Shi & Vakoc, 2014), enhancer of zeste homolog 2 (EZH2) catalyzes methylation at histone-3 lysine 27 (H3K27) leading to transcriptional repression (Ai et al, 2017; Sermer et al, 2019). EZH2 emerged in recent literature as a potential therapeutic target for cancers (Ezhkova

[1]Department of Surgery, School of Medicine, University of Virginia, Charlottesville, VA, USA [2]Department of Biomedical Informatics, College of Medicine, The Ohio State University, Columbus, OH, USA [3]Department of Surgery, College of Medicine, The Ohio State University, Columbus, OH, USA [4]Department of Molecular Physiology and Biological Physics, School of Medicine, University of Virginia, Charlottesville, VA, USA [5]Robert M Berne Cardiovascular Research Center, University of Virginia, Charlottesville, VA, USA

Correspondence: lg8zr@virginia.edu; bw2pw@virginia.edu
*Mengxue Zhang and Go Urabe contributed equally to this work.

et al, 2011; Ai et al, 2017). Moreover, pharmacological evidence from our (Zhang et al, 2017, 2020 *Preprint*) and others' studies (Liang et al, 2019; Lino Cardenas et al, 2019) supports an IH-mitigating effect of pan-EZH inhibition. However, the EZH isoform-specific role in IH and underlying epigenetic mechanisms remained little known. In an extrapolated perspective, the interplay of chromatin modulators such as EZH2 and BRD4 is overall under-studied, especially in the context of neointima and SMC pathophysiology.

A powerful approach to tackling these issues is chromatin immunoprecipitation coupled with high throughput sequencing (ChIPseq). However, ChIPseq epigenomic studies pertaining to IH have been limited, and mostly confined to cell cultures (Das et al, 2017; Yao et al, 2018; He et al, 2019) which as oversimplified systems provide incomplete or even inaccurate information for interpreting in vivo processes (Perisic Matic et al, 2016; Stratton et al, 2019). Here we performed ChIPseq using angioplasty-injured rat carotid arteries that underwent IH. We observed a prominent injured-versus-uninjured genome-wide upsurge of H3K27me3, a gene repression mark (Lavarone et al, 2019). This was initially counter-intuitive to us since massive gene activation has been regarded as the prevailing event that prompts neointimal hyperproliferation (Marx et al, 2011; Byrne et al, 2015). Further analysis revealed angioplasty-induced H3K27me3 peak redistribution to anti-proliferative genes from pro-proliferative genes, suggesting repression of the former and de-repression of the latter in keeping with the gene-repressing role of EZH2/H3K27me3. Furthermore, the levels of EZH2 and its catalytic product H3K27me3 were found to be controlled by BRD4 in vivo in a SMC-specific setting. As such, this study sheds new light on epigenetic regulations key to IH pathophysiology.

# Results

### Angioplasty induces a genome-wide surge of H3K27me3 occupancy in rat carotid arteries

There has been a paucity of knowledge on in vivo genome-wide epigenetic regulations during angioplasty-induced neointima formation. In vitro ChIPseq using cultured cells is technically more convenient, but important information of in vivo pathological processes would be inevitably missed. On the other hand, there is a well-established, highly reproducible model of IH, namely, balloon angioplasty of rat common carotid artery to mimic clinical angioplasty (Clowes et al, 1983; Wang et al, 2015, 2018). This model confers an opportunity for in vivo ChIPseq studies using arteries that undergo IH. Typically, the angioplasty procedure with an inflated balloon damages the artery wall and its endothelial inner lining, thereby exposing SMCs to the blood. Consequently, SMCs—the major constituent cell population in the artery wall—become abruptly subjected to various stimuli such as PDGFs, and they migrate and amplify, forming highly cellular neointimal lesions. We collected the arteries at post-angioplasty day 7, the peak time of a myriad of pro-IH molecular and cellular events such as pro-proliferative gene activation (Marx et al, 2011; Saito et al, 2011; Wang et al, 2015; Shi et al, 2019). In keeping with the literature and hence validating this IH-inducing model, NRP2 and UHRF1, two

recently reported novel IH-prompting factors (Pellet-Many et al, 2015; Elia et al, 2018), were both up-regulated in injured versus uninjured arteries at post-angioplasty day 7 (Fig S1). For ChIPseq, we used the balloon-angioplastied (denoted herein as injured) common carotid artery and its sham control from the same animal, which is the contralateral carotid artery that received surgery but not balloon angioplasty (denoted as uninjured).

To survey gene-activating chromatin remodeling, we performed ChIPseq using H3K27ac and BRD4 as chromatin marks that are associated with active enhancers and promoters (Anders et al, 2014; Brown et al, 2014). To monitor gene-repressing remodeling, we chose H3K27me3 for ChIPseq, which is a well-documented histone mark of gene repression as opposed to H3K27ac (Wassef et al, 2019). Fig 1A heat maps illustrate binding density of chromatin marks in a 10-kb swath flanking each transcription start site and hierarchical clustering of the ChIPseq peaks. It appears that in Cluster-1 and Cluster-2, the majority of the H3K27ac and H3K37me3 peaks are mutually exclusive. That is, Cluster-1 categorizes the peaks high in H3K27me3 signal and low in BRD4, H3K27Ac and H3K4me1; by contrast, Cluster-2 includes the peaks high in BRD4, H3K27Ac, and H3K4me1 but low in H3K37me3. The remainder peaks of low ChIPseq signal across experimental conditions fall in the category of Cluster-3. Gene annotation reveals that gene-regulatory factors are top scored in both Cluster-1 and Cluster-2 (Figs 1B and S2).

Distribution of individual ChIPseq peak scores is presented as bean plot in Fig 2A. Of note, H3K27ac peak scores increased overall in injured arteries compared to uninjured sham controls—a result fitting in the traditional view of injury-induced gene activation. The majority of the BRD4 ChIPseq peaks overlapped with that of H3K27ac (see Venn diagrams in Fig S3A–C), as expected since both are associated with active enhancers (Anders et al, 2014; Brown et al, 2014). Both BRD4 and H3K27ac peaks overlapped with that of H3K4me1. The total number of H3K4me1 peaks was greater, which is reasonable as H3K4me1 enriches not only at active enhancers but also inactive and poised enhancers (Brown et al, 2014). However, H3K27me3 peaks augmented markedly rather than abated after injury (*P*-value: $1.1 \times 10^{-288}$, Fig 2A). This was unexpected, since H3K27me3 is a gene repression mark (Doni Jayavelu et al, 2020) whereas gene activation has been traditionally deemed the predominant event after arterial injury (Marx et al, 2011). The post-injury gain of H3K27me3 peak coverage becomes particularly conspicuous in the scatter plot (Fig 2B); obviously, there is a prevailing shift to the upper side, i.e., increased H3K27me3 peak coverage in injured (versus uninjured) arteries. Thus, these results for the first time reveal prominent genome-wide surges of not only H3K27ac but also H3K27me3 occupancy in vivo in the angioplasty-induced IH model of rat carotid artery.

### H3K27me3 redistributes from pro-proliferative genes to anti-proliferative genes after angioplasty

To take a closer look at the enrichment of these epigenomic marks, we focused on *Cdkn1c* and *Ccnd1*, genes encoding P57 and cyclin-D1—bona fide anti-proliferative and pro-proliferative factors, respectively (Marx et al, 2011). As illustrated by Fig 3A and B (ChIPseq peak coverage presented in Table S1), H3K27me3 ChIPseq peaks intensified at *P57* but ebbed at *Ccnd1* after arterial injury; in

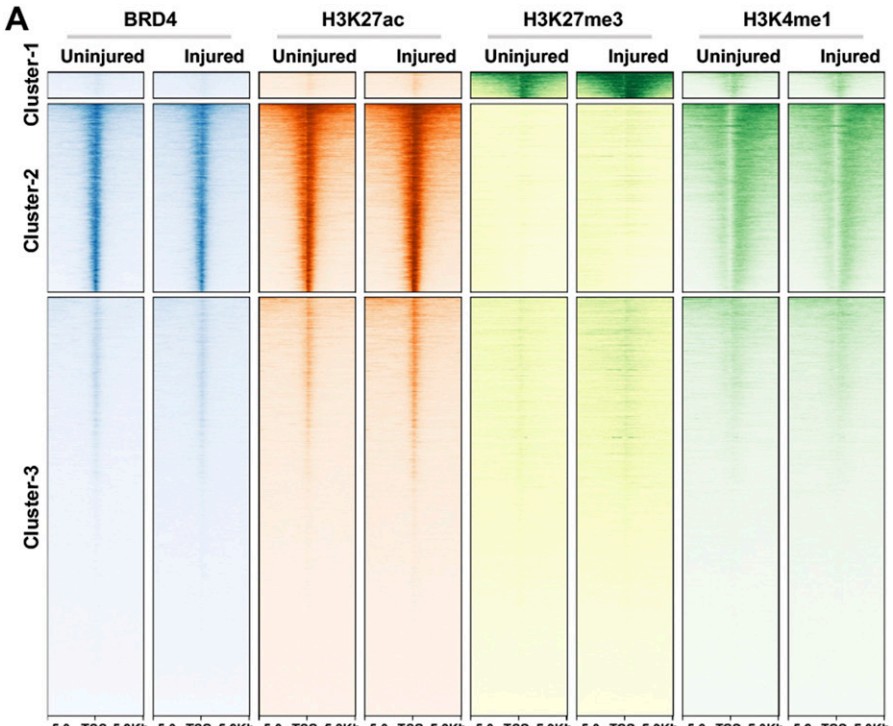

**A**

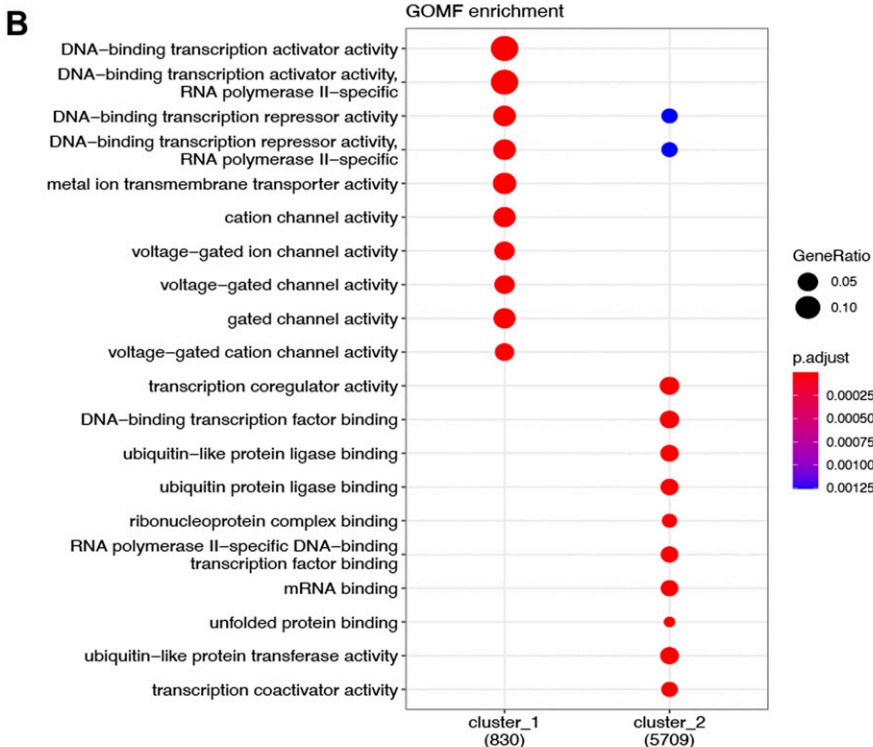

**B**

**Figure 1. Injury-induced genome-wide changes of ChIPseq reading in rat common carotid arteries.**
Balloon-injured rat left common carotid arteries and contralateral arteries (uninjured sham control) were collected (each group pooled from 50 rats) at day 7 post angioplasty and snap-frozen for use in ChIPseq experiments. **(A)** ChIPseq heat map showing binding density of BRD4, H3K27ac, H3K27me3, or H3K4me1. ChIPseq signal anchors a 10 kb center region with 5 kb flanking on either side of the transcription start site. Hierarchical clustering highlights injury-induced increase of H3K27me3 ChIPseq signal (Cluster-1) and non-overlap between H3K27me3 and H3K27ac. **(B)** Functional enrichment of Clusters 1 and 2. Presented is dot plot of Top 10 molecular function (MF) gene ontology (GO) terms with color as adjusted *P*-value and size of dot as gene ratio. Enrichment analysis was performed by clusterProfiler.

contrast, H3K27ac occupancy markedly increased instead at *Ccnd1*. Moreover, injury-induced H3K27me3-up/H3K27ac-down were found at the gene of BMP4 (Fig S4), an anti-proliferative factor in SMCs that counters IH (King et al, 2003; Corriere et al, 2008), and H3K27ac-up/

H3K27me3-down occurred at the gene loci of other pro-proliferative factors including UHRF1 (Fig 3C) and NRP2 (Fig S4), both recently reported to promote SMC and neointima proliferation (Pellet-Many et al, 2015; Elia et al, 2018). Thus, there appeared to be an injury-

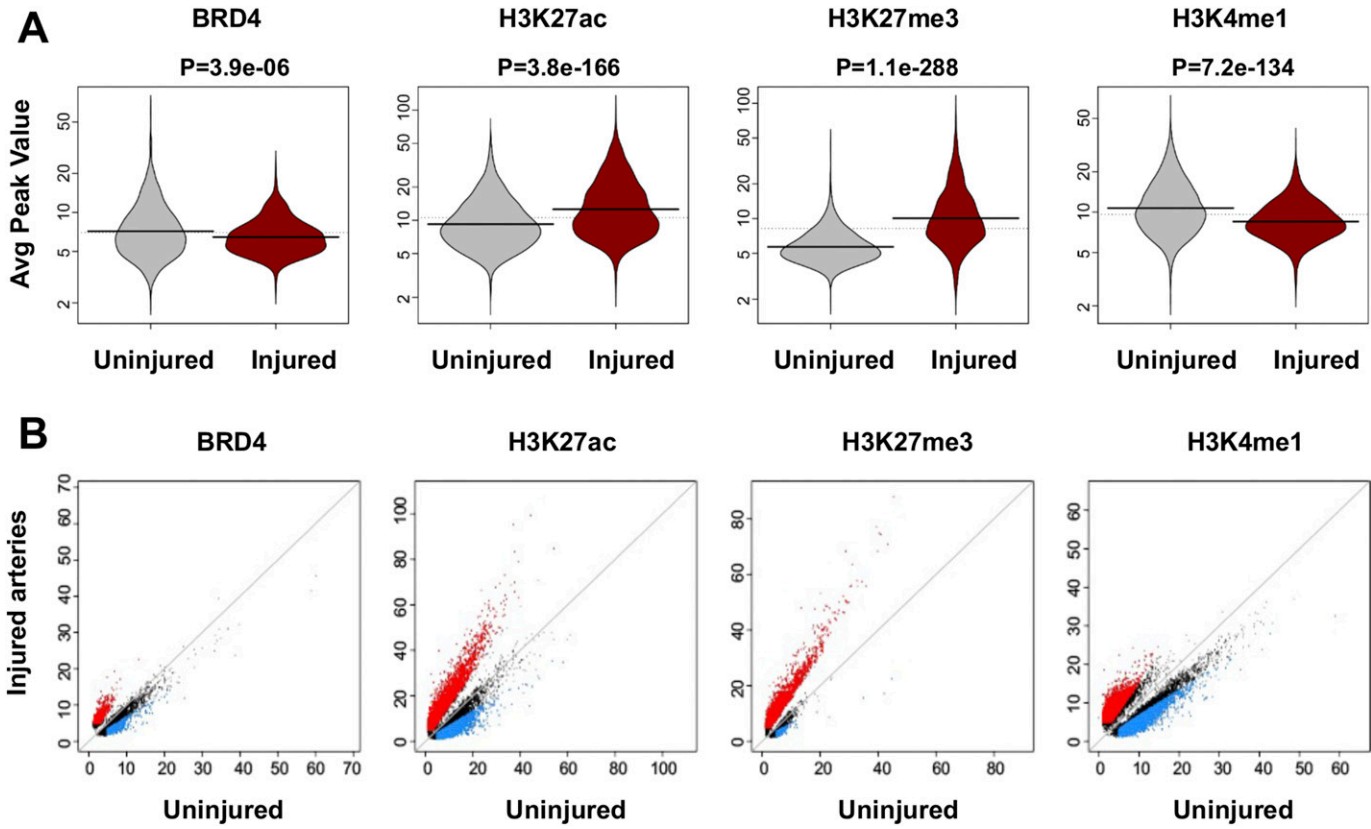

**Figure 2.  Comparison of ChIPseq peak coverage between injured and uninjured arteries.**
**(A)** Bean plot showing genome-wide distribution of BRD4 or histone mark ChIPseq peak values. *P*-values from Wilcox test are presented above the plots. **(B)** Scatter plot showing injury-induced change in binding density (ChIPseq reads) of BRD4 or a histone mark. Red and blue indicates increase and decrease, respectively, with a twofold cutoff.

induced redistribution of these opposing epigenomic marks; that is, increase of H3K27me3 and decrease of H3K27ac at anti-proliferative genes and the opposite at pro-proliferative genes (Fig 3D). It was further interesting to note that the majority of the ChIPseq peaks of active enhancer marks, H3K27ac and BRD4, largely aligned, and increased at *Ccnd1*, *Uhrf1*, and *Nrp2* after injury. Moreover, injury-induced H3K27ac/BRD4 co-enrichment occurred at enhancers not only in intronic regions (Figs 3 and S4) but also in the upstream of TSS far from a promoter (see *Ccnd1*, Fig 3B). Taken together, the foregoing results indicate angioplasty-induced H3K27me3 and H3K27ac genomic redistribution in rat carotid arteries—an observation not previously reported.

### H3K27me3 writer EZH2's expression is governed by H3K27ac reader BRD4 in SMCs

Given the striking angioplasty-induced H3K27me3 upsurge and redistribution, we were inspired to next investigate regulators of EZH2, the methyltransferase that deposits H3K27me3 (Wassef et al, 2019). To this end, ChIPseq data (Fig 4A and B) provided an interesting clue of greater BRD4/H3K27ac occupancy (injured versus uninjured) at enhancers in *Ezh2* intronic and upstream regions. We thus determined enhancer's importance for *Ezh2* expression by using two CRISPR approaches. We first performed enhancer-targeting genome editing through nuclease-active Cas9 and

observed reduction in EZH2 mRNA and protein (Fig 4C). In a negative control experiment, EZH2 mRNA was not reduced when we applied sgRNAs that targeted an upstream region very much away (50 kb) from the *Ezh2* TSS (Fig S5). Since genome editing possibly involves off-targets, we then applied a non-genome editing method (Kearns et al, 2014) (used in our recent report) (Yang et al, 2019) which is gaining popularity for minimizing off-target concerns. In principle, guided by sgRNA, deactivated (or dead) Cas9 fused with a repressor protein binds to the targeted enhancer (yet without cutting), thereby hindering its transcription-enhancing function. Using this approach, we again observed a decrease in EZH2 mRNA (sgRNA versus scrambled, Fig 4D). Furthermore, the experiments with siRNA-transfected SMCs indicated that BRD4, but not other BET family members (BRD2 and BRD3), was a determinant of EZH2 transcript and protein levels (Fig 4E and F). Thus, these results together suggest a BRD4/enhancer epigenetic control of EZH2 expression (Fig 4G) in accordance with the foregoing ChIPseq data.

### SMC-specific BRD4 deletion reduces EZH2 and H3K27me3 in injured mouse arteries

We next examined whether the BRD4→EZH2 regulatory axis also occurred in vivo. We first performed conditional KO of BRD4 and

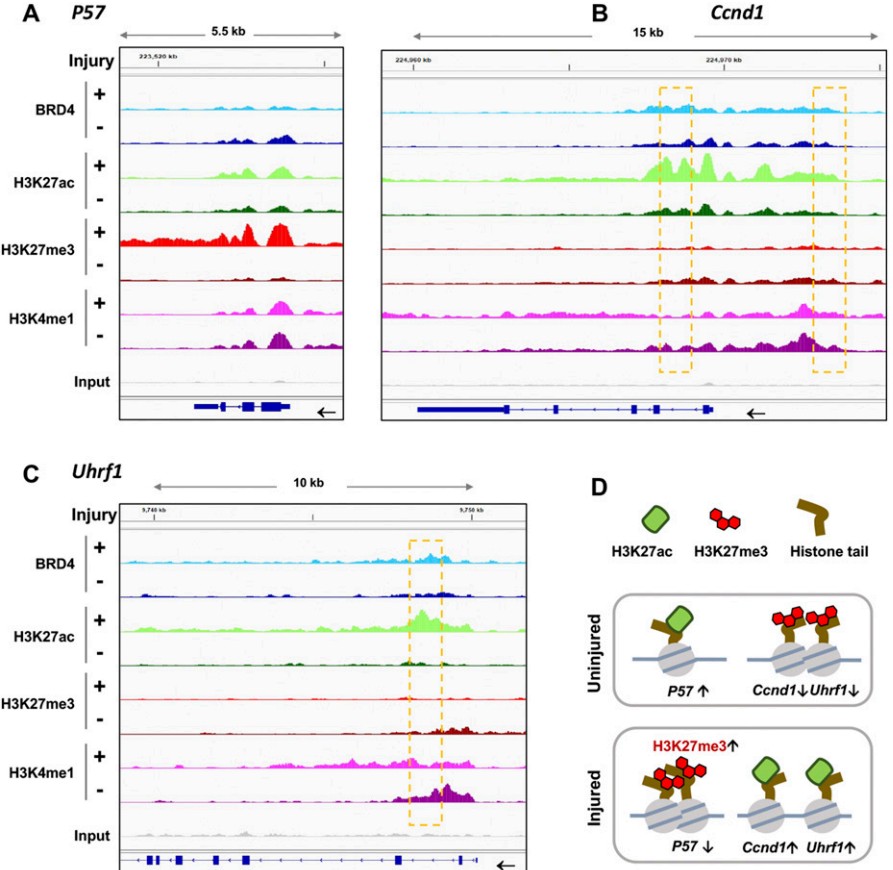

Figure 3. Injury-induced H3K27me3 enrichment at anti-proliferative gene P57 and BRD4/H3K27ac enrichment at pro-proliferative gene Ccnd1. ChIPseq was performed as described for Fig 1. Integrative genomics viewer (IGV) tracks show comparison of normalized ChIPseq peaks between injured arteries (+, light color) and uninjured sham-control arteries (−, dark color). Boxes highlight the regions where the binding of H3K27ac and BRD4 increased after arterial injury. Non-specific input confirms low background noise. **(A, B, C)** IGV profiles of ChIPseq peaks illustrating loci-specific binding density of chromatin marks. **(D)** Schematic proposition of H3K27me3 and H3K27ac redistribution between anti-proliferative (e.g., *P57*) and pro-proliferative genes (e.g., *Uhrf1* and *Ccnd1*), based on the artery tissue ChIPseq analysis. Note: The cartoon does not represent the true genomic locations of *Uhrf1* and *Ccnd1*.

IH-inducing wire injury (Fig 5A). Mice were cross-bred with the strains of *Brd4$^{fl/fl}$* and *Myh11-CreER$^{T2}$*. Tamoxifen-containing chow was fed to *Brd4$^{fl/fl}$; Myh11-CreER$^{T2}$* mice to induce SMC-specific BRD4 KO followed by wire injury and collection of femoral arteries for histology (Fig 5B). Immunostaining confirmed tamoxifen-induced BRD4 KO (Fig 5C). Interestingly, as seen in Fig 5D and E, IH (measured as I/M ratio) was drastically reduced in homozygous BRD4 KO mice, either compared with the wild-type (*Brd4$^{+/+}$*) or heterozygous (*Brd4$^{+/−}$*) animals. This concurred with our previous reports using pharmacological and shRNA approaches (Wang et al, 2015; Zhang et al, 2019). Thus, this result from SMC-specific BRD4 KO mice is significant because it represents the first time demonstration of the SMC-specific role of BRD4 in angioplasty-induced neointimal development.

We then measured on artery cross sections the levels of EZH2 (Fig 6A and B) and its catalytic product H3K27me3 (Fig 6C and D). Immunofluorescence indicated that BRD4 KO substantially reduced EZH2 in the neointimal layer, and there was a clear trend of reduced EZH2 protein when quantified in the media or neointima/media combined although the changes did not reach statistical significance. Accordingly, the level of H3K27me3 was markedly lower in *Brd4$^{fl/fl}$; Myh11-CreER$^{T2}$* versus *Brd4$^{fl/fl}$* mice, as observed in media/neointima layers, with changes nearly significant if quantified separately in the neointima or media. Thus, BRD4 controls EZH2 expression and

hence H3K27me3 levels in vivo in injured mouse femoral arteries.

## EZH2 and EZH1 each promotes IH in angioplasty-injured rat carotid arteries

Inasmuch as BRD4 is a determinant of EZH2 expression and IH, as found herein, we inferred that EZH2 would play a positive role in IH as well. Consistently, EZH2 protein was up-regulated in rat carotid arteries after angioplasty (Fig S6A and B). However, there are two EZH isoforms and their isoform-specific role in IH was not previously differentiated. Compared with EZH2, EZH1 is a much less studied isoform with no known IH-associated function. To determine the specific roles of the two EZH isoforms in neointimal development, we performed gain-of-function experiments taking advantage of the method of local lentiviral gene transfer to the injured artery wall (Huang et al, 2020) (Fig 7A). We found that compared with the GFP control, increasing EZH2 heightened H3K27me3 (Fig 7B) and exacerbated IH and restenosis (lumen narrowing) (Fig 7C). Consistently, immunostained mitotic marker proliferating cell nuclear antigen overlapped with H3K27me3 in the nuclei of periluminal neointimal cells (Fig S7). One would thus predict that blocking the enzymatic activities of both EZH1 and EZH2 should be effective for IH inhibition. Indeed, IH diminished in the loss-of-function experiment with the pan-EZH1/2 inhibitor UNC1999 applied (Fig 7D and E). This result concurred

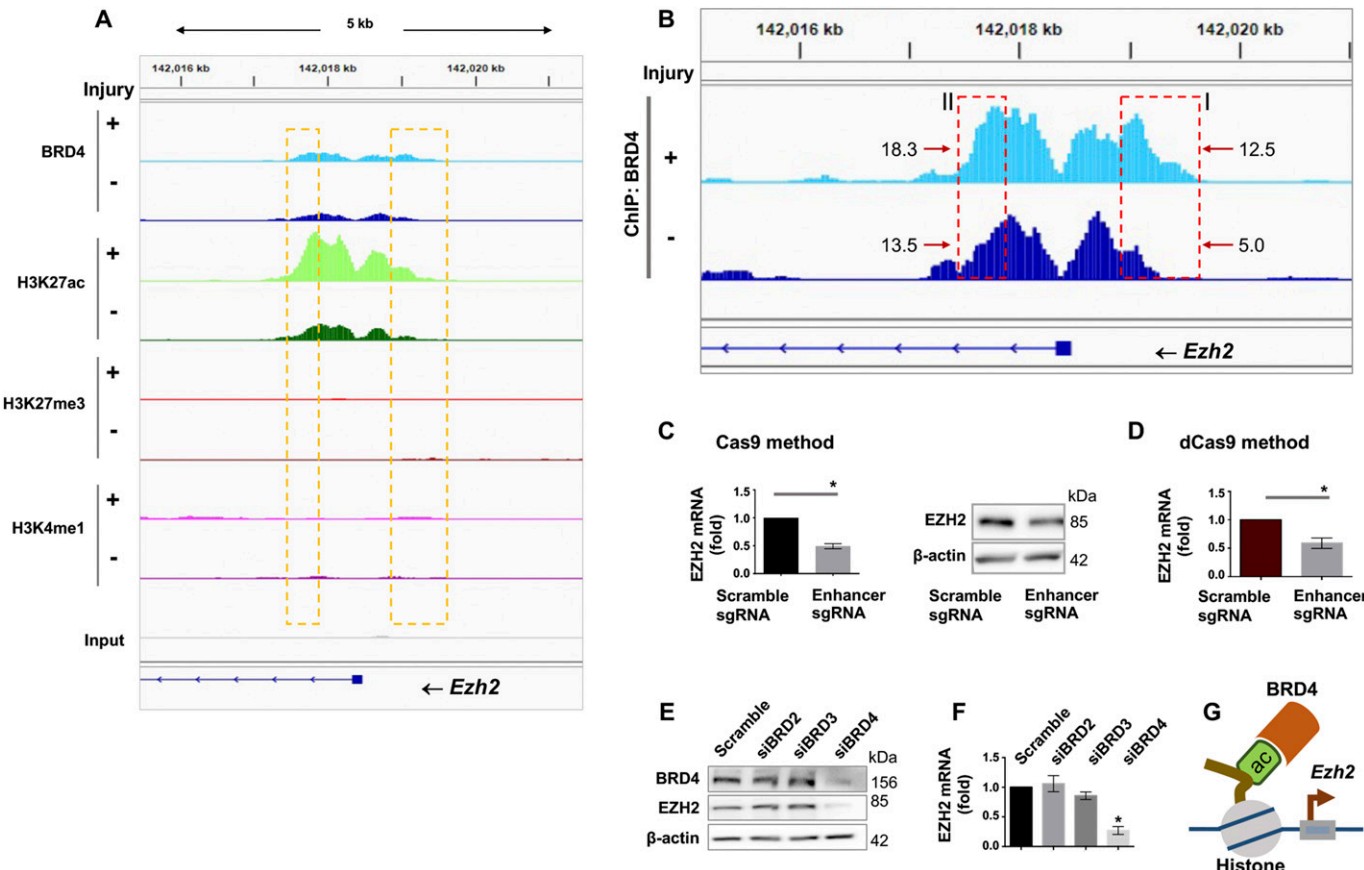

**Figure 4. Regulation of EZH2 expression by BRD4 in smooth muscle cells (SMCs) in vitro.**
**(A, B)** BRD4 ChIPseq peaks at *Ezh2* (B is the zoom-in version of A). Boxes highlight one enhancer region at the distal end of *Ezh2* promoter and another within an intron. Genomic coordinates of Box-I: chr4 142018866-142019623; Box-II: chr4 142017383-142017905. Mean coverage of ChIPseq peaks (indicative of BRD4 binding dencity) within each box is labeled, with an arrow pointing to the box. Note the H3K4me1 signal is low relative to H3K27ac of the same scale. **(C)** Effect of Cas9-mediated enhancer deletion (genome editing) on EZH2 expression in rat primary aortic SMCs. sg, small guide RNA. The pair of sgRNAs flank an *Ezh2* intronic region, that is, +1,161 bp to +218 bp from the transcription start site. Quantification: Mean ± SEM; n = 3 independent experiments; paired *t* test, *P < 0.05. **(D)** Disruption of EZH2 expression through a non-genome-editing, dead Cas9 (dCas9)-facilitated approach. Rat primary aortic SMCs were used. The same sgRNAs (as that in C) were used. Quantification: Mean ± SEM; n = 3 independent experiments; paired *t* test, *P < 0.05. **(E, F)** Effect of BRD4 silencing on EZH2 expression. BRD2, BRD3, or BRD4 was silenced with their specific siRNAs (validated in our recent reports) (Wang et al, 2015; Zhang et al, 2019). Cultured rat aortic SMCs were starved for 6 h before transfection with the siRNA for BRD2, 3, or 4 overnight. The cells recovered (without transfection reagents) for 24 and 48 h before RNA and protein extraction, respectively. EZH2 protein and mRNA were measured with Western blot and qRT-PCR (normalized by ΔΔCT-log₂) assays. Quantification: Mean ± SEM; n = 3 independent experiments; one-way ANOVA with Bonferroni test, *P < 0.05 compared with the scrambled-sequence siRNA control. **(G)** Schematic depicting BRD4 and its co-localization with H3K27ac at *Ezh2* that promote *Ezh2* transcription.
Source data are available for this figure.

with other lines of pharmacological evidence (Liang et al, 2019; Lino Cardenas et al, 2019). Pharmacological observations are inevitably confounded by off-target effects. Moreover, pan-EZH inhibitors cannot distinguish between EZH2 and EZH1 especially when applied in vivo where it is impractical to control inhibitor concentrations in tissues. In this regard, our findings made through EZH1- or EZH2-specific expression are important, which not only delineate a pre-conceived pro-IH function of EZH2 (Liang et al, 2019; Lino Cardenas et al, 2019) but also uncover a previously unknown in vivo role for EZH1 in promoting IH.

### EZH2 and EZH1 each promotes SMC proliferation and migration in vitro

To dissect EZH-mediated functional mechanisms, we used the PDGF-induced cellular model that exhibits salient pro-IH migratory and proliferative SMC phenotypic transition (Wang et al, 2015).

Pretreatment with the pan-EZH1/2 inhibitor UNC1999 concentration-dependently inhibited PDGF-induced SMC proliferation and migration (Fig S8A and B). Furthermore, in an isoform-specific manner, silencing either EZH2 or EZH1 with shRNA markedly inhibited PDGF-induced SMC proliferation and migration (Fig 8A–D). It is common that some isomers are functionally redundant; in other words, knocking down/out one isomer fails to impart a significant functional effect because of the compensatory effect of other isomers(s). Opposing this scenario, our results indicated that EZH2 and EZH1 were non-redundant in promoting the pro-IH SMC behaviors. This is interesting, given that redundancy of EZH2 and EZH1 was reported in other biological contexts (Ezhkova et al, 2011; Wassef et al, 2019). In further support of this conclusion, lentivirus-mediated gain-of-function experiments indicated that increasing either EZH2 or EZH1 enhanced SMC proliferation and migration (Fig 8E–H) and exacerbated IH (Fig 7). Thus, these

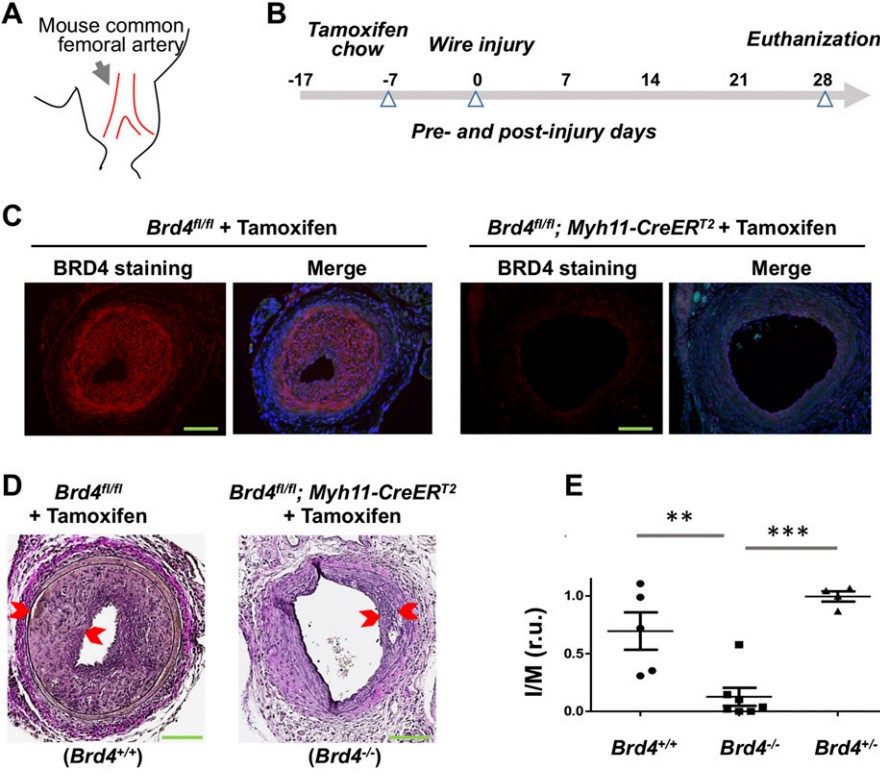

**Figure 5. Smooth muscle cell-specific BRD4 KO reduces IH in wire-injured mouse femoral arteries.**
**(A)** Cartoon of mouse common femoral artery where wire injury was made to induce IH. **(B)** Diagram indicating the time line for tamoxifen feeding, wire injury, and tissue collection. **(C)** Immunofluorescence confirming tamoxifen-induced BRD4 KO in mouse arteries. Scale bar: 50 $\mu$m. **(D)** Comparison of IH between WT ($Brd4^{fl/fl}$) and smooth muscle cell-specific BRD4 KO ($Brd4^{fl/fl}$; $Myh11$-$CreER^{T2}$) mice. Neointima thickness is demarcated by arrow heads. IH is normalized as intima/media area (I/M) ratio. Scale bar: 50 $\mu$m. **(E)** Quantification: Mean ± SEM; n = 4–7 mice, as indicated by the data points in scatter plots. Statistics: one-way ANOVA with Bonferroni test; **$P$ < 0.01, ***$P$ < 0.001; r.u., relative unit.

in vitro results together with the foregoing in vivo evidence demonstrate that EZH2 and EZH1 each plays a positive role in proliferative/migratory SMC phenotypic transition and neo-intimal development.

### EZH2 and EZH1 each regulates the expression of both P57 and cyclin-D1

To better understand the profound impact of EZH2 on SMC proliferation/migration, we next investigated its possible down-stream molecular effectors. We first looked into P57 and cyclin-D1, the well-documented representative anti-proliferative and pro-proliferative factors, respectively (Marx et al, 2011). Pretreat-ment of SMCs with UNC1999 partially de-repressed $P57$ expression and inhibited $Ccnd1$ expression, at both mRNA and protein levels in the presence of PDGF-BB (Fig 9A and B). The data from shRNA expression demonstrated that either EZH2 or EZH1 loss-of-function partially reinstated PDGF-suppressed $P57$ expression (no significance in the case of EZH1) while blocking the PDGF induction of $Ccnd1$ expression (Fig 9D and E). Accordingly, the gain-of-function experiments led to an opposite result (Fig 9G and H). Although the impact of EZH1 overexpression on Cyclin-D1 mRNA and P57 protein, and that of EZH2 overexpression on P57 protein did not reach a statistical significance, the latter agreed with the literature evidence from a different cell type (Sermer et al, 2019). Nonetheless, multiple lines of evidence indicate that EZH2 and EZH1 each negatively regulates anti-proliferative P57 and positively regulates pro-proliferative Cyclin-D1 in SMCs in vitro.

### EZH2 regulates the expression of UHRF1, another pro-IH chromatin modulator

In the quest for novel targets that responded to angioplasty-induced EZH2/H3K27me3 up-regulation, UHRF1 appeared as an attractive candidate based on the ChIPseq data. As seen in Fig 3C (boxed region), H3K27ac peaks rose whereas H3K27m3 peaks de-clined at $Uhrf1$ after injury, implicating $Uhrf1$ activation. Recently, UHRF1 was functionally linked to the reading of both histone methylation and acetylation (Taniue et al, 2020). Furthermore, $Ezh2$ and $Uhrf1$ were found in the same gene network both promoting keratinocyte self-renewal (Mulder et al, 2012). However, whether EZH2 regulates UHRF1 expression was not known. Here we found that UNC1999 blocked PDGF-stimulated SMC $Uhrf1$ mRNA expres-sion (Fig 9C). While EZH2 (or EZH1) loss-of-function reduced, their gain-of-function increased $Uhrf1$ transcripts (Fig 9F and I). Thus, we identified $Uhrf1$ as a target gene of the H3K27me3 writer EZH2, a finding consistent with the recently reported positive role of UHRF1 in SMC proliferation and injury-induced IH (Elia et al, 2018). Im-portantly, guided by the in vivo ChIPseq data (Figs 3 and 4), the experiments of ChIP-qPCR using the in vitro SMC model demon-strated mitogen-induced H3K27ac enrichment at $Ezh2$ and $Uhrf1$ (Fig 9J), bridging in vitro and in vivo observations. Indeed, while UHRF1 markedly decreased in the neointima due to BRD4 KO in mouse arteries (Fig 10A and B), it was increased by EZH2 (or EZH1) gain-of-function in injured rat arteries (Figs 10C and D and S9). Collec-tively, these and other results in this study for the first time provide in vitro/in vivo evidence for a BRD4→EZH2→UHRF1 epige-netic signaling cascade. The new information helps mechanistic

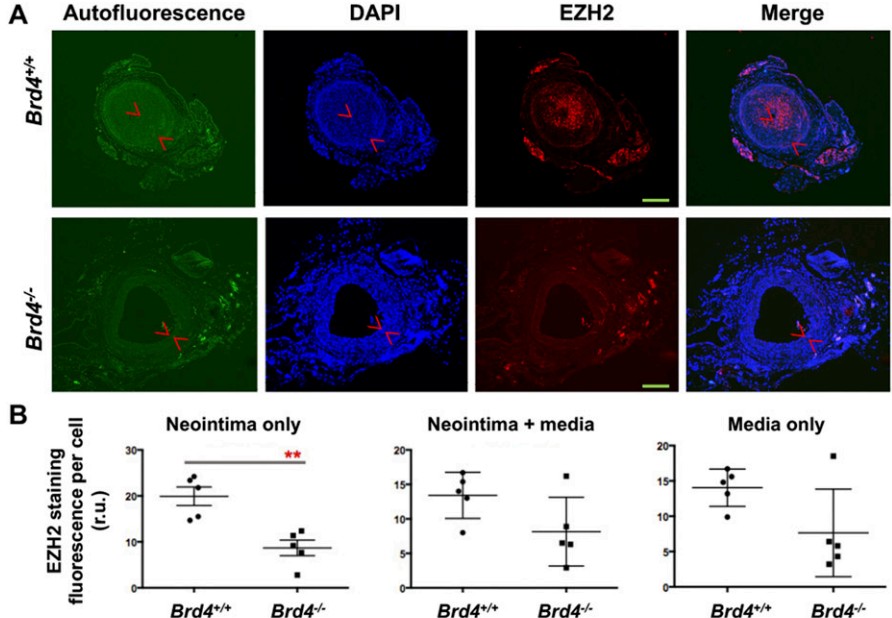

**Figure 6. Reduced EZH2 and H3K27me3 in mouse arteries of smooth muscle cell-specific BRD4 KO.**
Tamoxifen-induced BRD4 KO and wire injury were performed as described for Fig 5. **(A, B, C, D)** Immunofluorescence shows comparison of EZH2 (A, B) or its catalytic product H3K27me3 (C, D) between WT and BRD4 conditional KO mice. Neointima is demarcated by arrow heads. Fluorescence intensity was normalized to cell number (DAPI-stained nuclei). Scale bar: 50 $\mu m$. Quantification: Mean ± SEM; n = 5 mice as indicated by the data points in scatter plots. Statistics: nonparametric Mann–Whitney test following Shapiro–Wilk normality determination, *$P < 0.05$, **$P < 0.01$; r.u., relative unit.

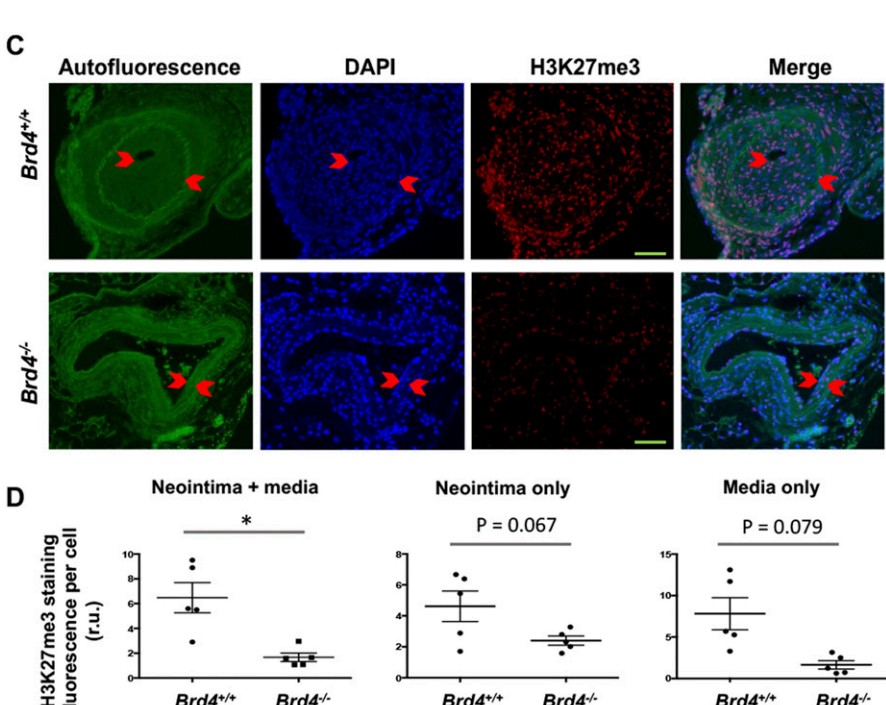

interpretation for the angioplasty-induced H3K27me3 upsurge and redistribution that occur during neointima formation.

## Discussion

Whereas epigenetics is increasingly recognized as crucial in cardiovascular diseases (Stratton et al, 2019), epigenome-scale studies pertaining to IH have been mostly reliant on cultured cells (Brown et al, 2014; Yao et al, 2018; He et al, 2019). Moreover, the nuanced mechanisms involving the interplay of chromatin modulators are overall poorly understood. We report here the first in vivo ChIPseq epigenomic survey focusing on angioplasty-induced IH. We found surging rather than declining H3K27me3 binding intensity after angioplasty. Further analyses revealed that whereas post-angioplasty H3K27ac occupancy increased at pro-proliferative genes (e.g., *Uhrf1*), H3K27me3 enrichment redistributed from these genes to anti-proliferative genes such as *P57*. Moreover, H3K27ac reader BRD4 dictated the level of the

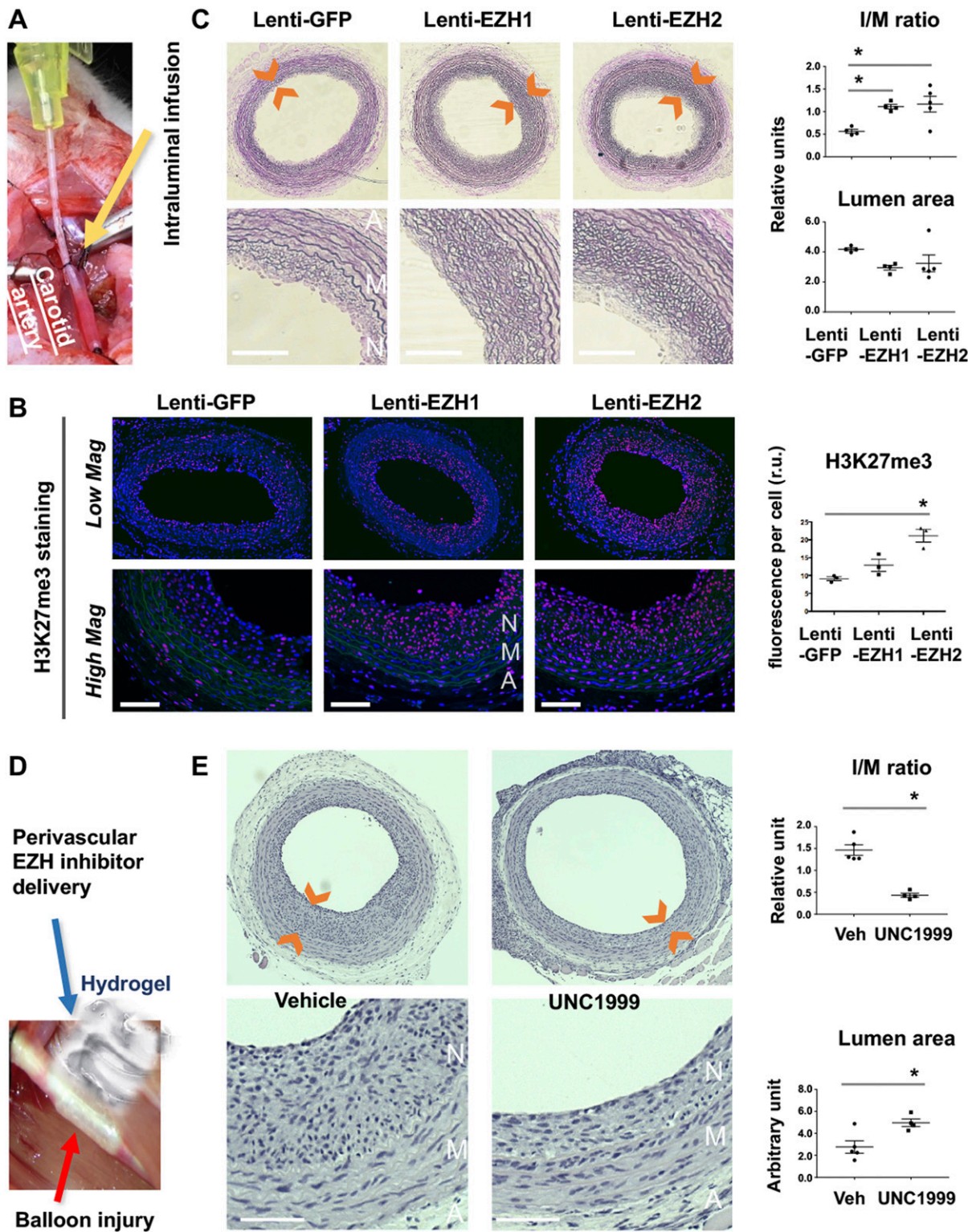

**Figure 7. Effect of EZH1 or EZH2 gain- or loss-of-function on IH in balloon-injured rat carotid arteries.**

**(A)** Picture illustrating intraluminal infusion of lentivirus to express a gene in the balloon-injured rat carotid artery wall. A cannula (yellow device) connected to a syringe was used to inject lentivirus to the carotid artery lumen for infusion into the injured artery wall. **(B)** Gain of function. EZH overexpression (EZH1 or EZH2 each in fusion with GFP) in the injured rat carotid artery wall was accomplished via intraluminal infusion of lentivirus. Arteries were collected at post-injury day 14 for histology. Immunostained cross sections indicate increase of H3K27me3 in EZH-overexpressing arteries versus Lenti-GFP controls, predominantly in the neointima layer. A, adventitia. M, media. N, neointima. Scale bar: 50 $\mu$m. Quantification: Mean ± SEM; n = 3 rats; one-way ANOVA with Bonferroni test, *P < 0.05. No significance between Lenti-EZH1 and Lenti-GFP. **(C)** EZH gain-of-function exacerbates IH (measured as I/M ratio). Neointima is demarcated between arrow heads. Scale bar: 50 $\mu$m. Quantification:

H3K27me3 writer EZH2, which in turn positively regulated UHRF1 expression. As such, these results illustrate previously unrecognized genome-wide H3K27me3 remodeling that occurs upon neointima formation and involves the interplay of epigenetic regulators BRD4, EZH2, and UHRF1.

Balloon angioplasty is a common procedure for treating cardiovascular diseases, predominantly atherosclerosis—a stenotic disease. Paradoxically, this treatment precipitates the formation of lumen-occupying neointima (so called IH), ultimately leading to recurrent stenosis. Angioplasty mechanically injures the artery and the endothelial lining of the inner vessel surface. This exposes medial SMCs to a myriad of stimuli in the blood that trigger proliferation/migration of SMCs which then build up neointima (Wang et al, 2018). Accordingly, the main theme of research and findings in regard to SMC/neointima proliferation has long been the activation of pro-proliferative/migratory genes and pathways (Marx et al, 2011; Byrne et al, 2015). In fact, studies on gene regulation in general have been dominated by a focus on gene activation rather than repression (Doni Jayavelu et al, 2020; Pang & Snyder, 2020). It was thus initially counter-intuitive for us to see an angioplasty-induced prevailing increase of genome-wide H3K27me3 binding intensity. Although pharmacological evidence exists for an IH-mitigating effect of pan-EZH inhibitors that hinder H3K27me3 deposition (Zhang et al, 2017; Liang et al, 2019; Lino Cardenas et al, 2019), to the best of our knowledge, there was a lack of in vivo IH-associated epigenome-scale study of H3K27me3 remodeling. Rather than merely showing a gross change of total methylated histone proteins (e.g., immunoblotting or staining), the current ChIPseq study allowed for elaboration of the binding intensity of histone marks (e.g., H3K27me3) at specific genomic loci. Along this line, it is important to note that although still underappreciated, recent progress contends that just as critical in biology is the process of gene repression (Doni Jayavelu et al, 2020; Pang & Snyder, 2020). In this perspective, the herein observed genome-wide surge of H3K27me3, a gene repression signal in a pro-proliferative in vivo model, is not only informative for vascular studies but also inspiring in regard to future research in chromatin biology.

It was further intriguing to observe angioplasty-induced redistribution of H3K27me3 in the genomic landscape. A working hypothesis emerges from the data of UHRF1 and cyclin-D1 (*Ccnd1*) which exemplify pro-proliferative factors, and from that of P57 (*Cdkn1c*), a bona fide representative of anti-proliferative factors (Marx et al, 2011). As schematized in Fig 3D, after arterial injury, H3K27me3 accumulates at *P57* to repress its transcription; however, at *Uhrf1* and *Ccnd1*, H3K27me3 diminishes, thereby de-repressing their transcription, whereas these sites are replenished instead with transcription-activating H3K27ac. While each of these changes favors SMC/neointima proliferation, their combination could impart a "double whammy" effect. In accordance with the post-angioplasty H3K27me3/H3K27ac remodeling, angioplasty up-regulates the H3K27me3 writer EZH2 (Ai et al, 2017); indeed,

H3K27ac reader BRD4 which governs EZH2 expression is also up-regulated (Wang et al, 2015). In an overview of the reconstructed epigenomic landscape, a pathway is likely hereby paved, leading to heightened SMC proliferation and IH.

Indeed, we found that BRD4 and H3K27ac, both active enhancer marks promoting transcription (Ozer et al, 2018). Shi and Vakoc (2014) and Bradner et al (2017), co-localized at *Ezh2*, and their occupancy increased after angioplasty. Accordingly, BRD4 dictated EZH2 expression in vitro and in vivo. In line with our finding, enhanced efficacy by combining BRD4 and EZH2 inhibitors has been reported in oncology (Huang et al, 2018). In support of the IH-associated importance of this BRD4/EZH2 axis, reports indicated that BRD4 dramatically increased in the neointima and it was implicated as a driver of IH in a model induced either by angioplasty (Wang et al, 2015) or vein grafting (Zhang et al, 2019) although a SMC-specific role for BRD4 was not determined. Herein, our study demonstrated an IH-promoting function of BRD4 using mice of SMC-specific BRD4 KO. A pro-IH effect of EZH2 gain-of-function was also observed, in keeping with a BRD4→EZH2 hierarchical relationship. In addition, our data for the first time indicated non-redundancy of EZH2 and EZH1 in SMC proliferation/migration and IH. Functional redundancy of isoforms manifests when the silencing of one isoform is compensated for by the other isoform(s) and hence would not produce a covert effect. However, there was no obvious compensatory effect in either EZH1 or EZH2 loss- or gain-of-function experiments. This non-redundancy was somewhat a surprise because EZH1 was deemed redundant to EZH2 in other tissues and diseases, for example, skin and tumor (Ezhkova et al, 2011; Wassef et al, 2019). Moreover, EZH1 is a much less studied isoform and its function was not known in SMC pathophysiology, particularly in IH. Echoing our result of non-redundancy between EZH1 and EZH2, their differentiated roles were recently found in heart development and regeneration (Ai et al, 2017).

In the pursuit of effector genes downstream of EZH2, we were attracted to *Uhrf1*, where H3K27ac/BRD4 binding intensities increased whereas H3K27me3 binding decreased in injured (versus uninjured) arteries. Very recently, UHRF1 was reported to be a pro-IH factor (Elia et al, 2018). In another study, UHRF1 acted as a multifunctional epigenetic reader located at both histone methylation and acetylation marks (Taniue et al, 2020). UHRF1 and EZH2 were previously linked for their paralleled functions in keratinocyte self-renewal (Ezhkova et al, 2009) and for their positive correlation in human prostate tumor samples (Babbio et al, 2012). However, whether EZH2 and/or BRD4 impose an epigenetic control over UHRF1 was not known. Our ChIPseq data illustrated injury-stimulated BRD4/H3K27ac enrichment at *Uhrf1*. Moreover, while BRD4 KO reduced UHRF1 in injured mouse arteries, increasing EZH2 in injured rat arteries elevated UHRF. Thus, these and other results obtained herein are consistent with an epigenetic signaling axis of BRD4→EZH2→UHRF1. Epigenetic players have become increasingly appreciated for their importance in vascular homeostasis and dysregulation, yet their functional relationships

Mean ± SEM; n = 4–5 rats; one-way ANOVA with Bonferroni test, *$P$ < 0.05. **(D)** Picture depicting perivascular application of pan-EZH inhibitor UNC1999 dispersed in a thermosensitive hydrogel. **(E)** EZH loss of function (inhibition) mitigates IH. Arteries were collected at post-injury day 14. Scale bar: 50 $\mu$m. Quantification: Mean ± SEM; n = 4–5 rats; unpaired $t$ test, *$P$ < 0.05.

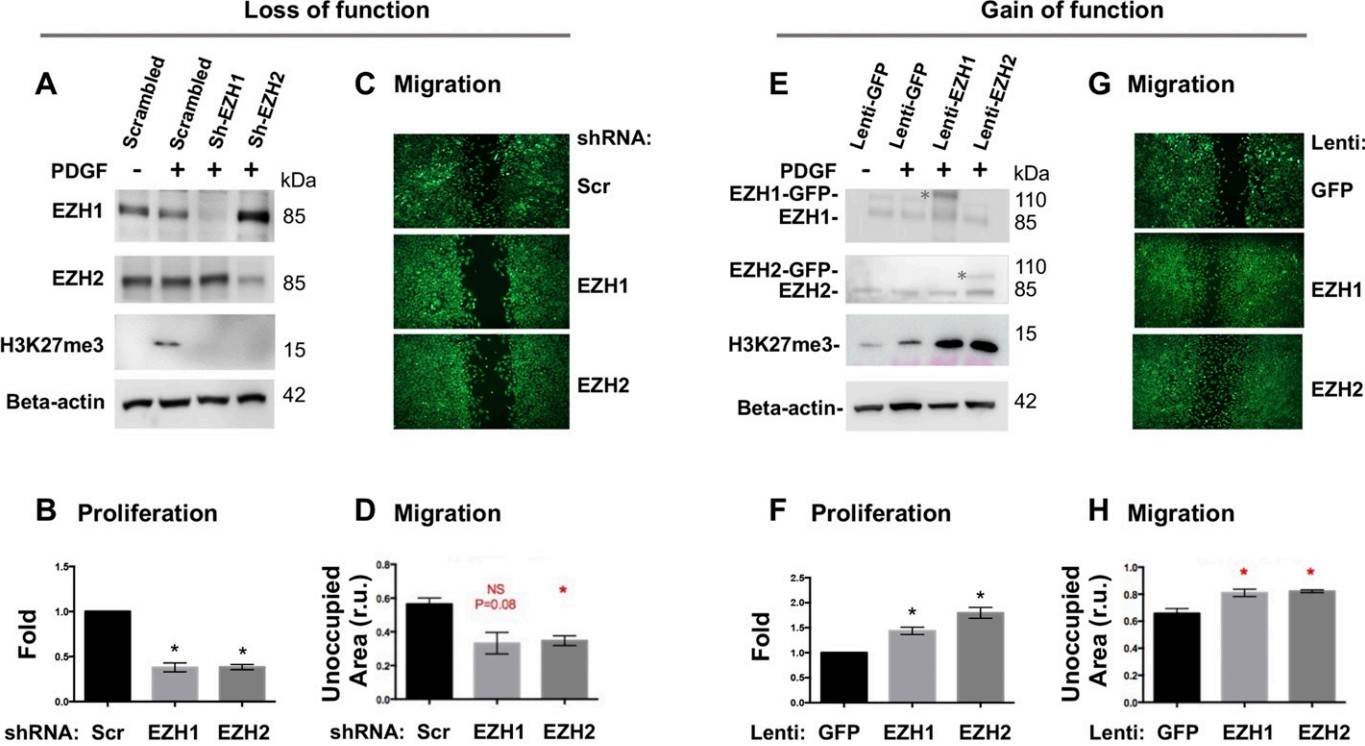

**Figure 8. Effect of EZH1 or EZH2 gain- or loss-of-function on smooth muscle cell proliferation and migration.**
**(A, B, C, D)** Loss of function. Silencing efficiency is indicated by Western blots (A). For proliferation (B) and migration (C, D) assays, cells were harvested at 72 h or imaged at 24 h after PDGF-BB stimulation, respectively. Scr, scrambled. NS, not significant. **(E, F, G, H)** Gain of function. EZH1 or EZH2 overexpression is shown in Western blots (E). * marks the recombinant protein in fusion with GFP; the lower band is endogenous protein. For proliferation (F) and migration (G, H) assays, cells were harvested at 72 h or imaged at 24 h after PDGF-BB stimulation, respectively. Quantification: Mean ± SEM; n = 3 independent experiments. Statistics: one-way ANOVA with Bonferroni test, *$P < 0.05$.
Source data are available for this figure.

remain poorly interpreted. To this end, herein we identified angioplasty-induced genome-wide chromatin remodeling that entails a BRD4→EZH2→UHRF1 regulatory cascade. It is important to further study this hierarchical (or possibly also co-operative) mechanistic axis and other involved chromatin-associated players, for better understanding of pro-IH molecular targets and the underlying chromatin biology.

## Conclusions

The current study presents integrated information from in vivo epigenome-scale survey, conditional KO and gene transfer, and the functional relation between chromatin regulators in rodent models of IH. The data revealed an angioplasty-triggered surge of genome-wide occupancy by H3K27me3, a gene repression mark. Moreover, H3K27me3 enrichment shifted to anti-proliferative genes from pro-proliferative genes where gene activation mark H3K27ac accumulated instead. In accordance, H3K27ac reader BRD4 enriched at the locus of the H3K27me3 writer EZH2 and governed its expression. These results highlight previously under-appreciated H3K27me3 remodeling that occurs upon neointima proliferation and entails a BRD4→EZH2→UHRF1 regulatory cascade. However, with future translation in mind, it is important to note limitations in the current study. These include the use of healthy animals without human-like

disease backgrounds, uncertain contribution of ECs in the ChIPseq samples, and the lack of data from human samples. pharmaceutical development of "epi-drugs" is rapidly advancing, though mainly in the cancer field (Shin & Bayarsaihan, 2017). To seize this momentum for improved treatments of vascular diseases, more research is warranted to delineate the specific roles of various chromatin modulators and relationships thereof of in the IH disease background.

# Materials and Methods

## Animals

All animal studies conform to the Guide for the Care and Use of Laboratory Animals (National Institutes of Health) and protocols approved by the Institutional Animal Care and Use Committee at University of Virginia.

## Balloon angioplasty in rat carotid arteries

To induce IH, the Fogarty balloon catheter for clinical thrombectomy (2F, Edwards Scientific) was applied in male Sprague–Dawley rats (300–350 g) to injure the left common carotid artery, as we previously described (Wang et al, 2015). The contralateral right

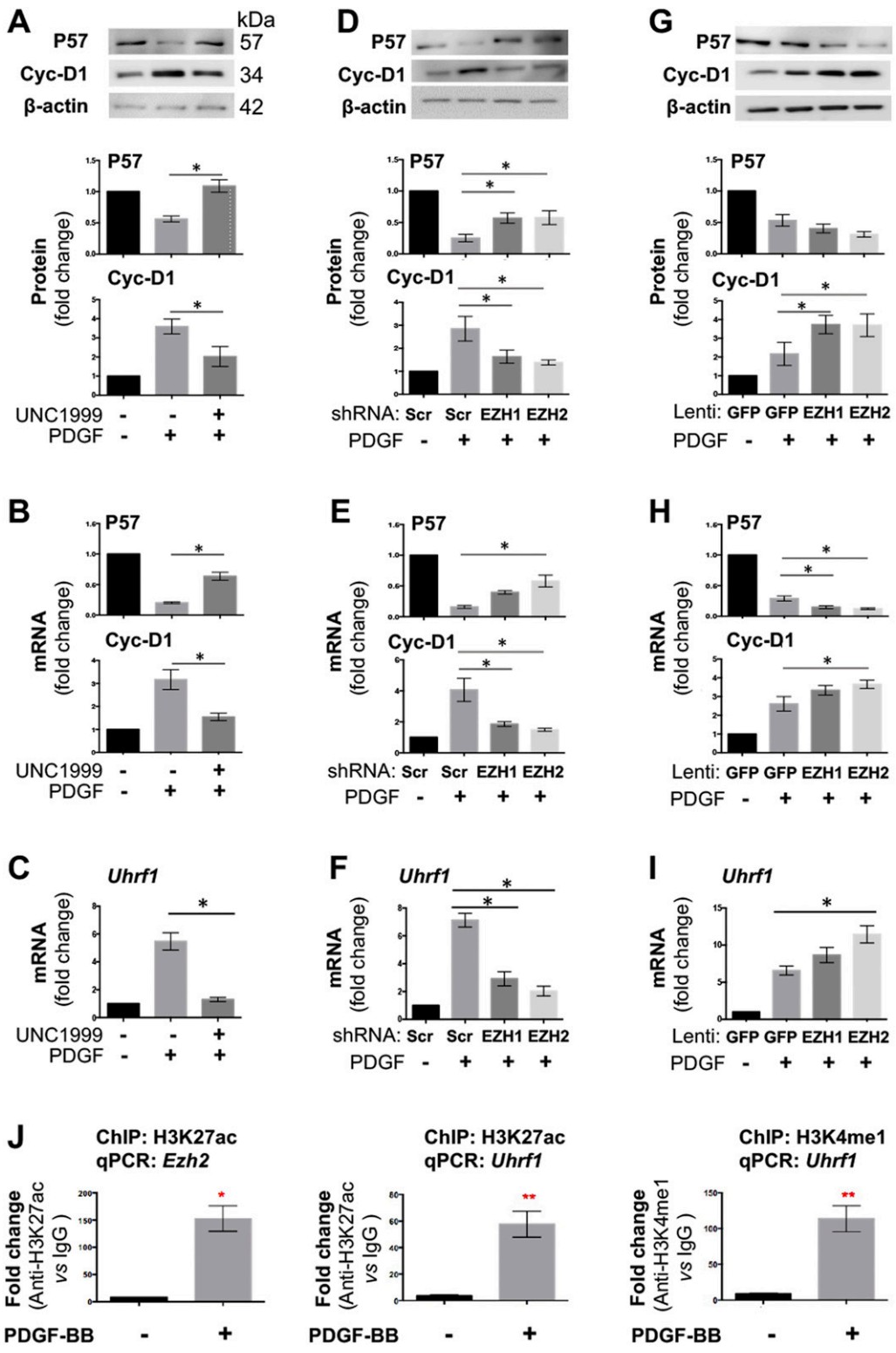

**Figure 9. Effect of EZH1 or EZH2 gain- or loss-of-function on target gene expression.**
MOVAS cells were pretreated with the pan-EZH1/2 inhibitor UNC1999 (5 $\mu$M) for 2 h, or transduced with lentivirus to silence or overexpress EZH1 or EZH2. Starved cells were stimulated with PDGF-BB (final 20 ng/ml) for 24 or 48 h before harvest for qRT-PCR or Western blot assay, respectively. Quantification: Mean ± SEM; n = 3 independent experiments. Statistics: one-way ANOVA with Bonferroni test, *$P$ < 0.05, **$P$ < 0.01. **(A, B, C)** Effect of pan-EZH1/2 inhibition on the expression of P57, cyclin-D1, and UHRF1. **(D, E, F)** Effect of EZH1 or EZH2 silencing on the expression of P57, cyclin-D1, and UHRF1. **(G, H, I)** Effect of increasing EZH1 or EZH2 on the expression of P57, cyclin-D1, and UHRF1. **(J)** ChIP-qPCR indicating H3K27ac or H3K4me1 binding at *Ezh2 or Uhrf1*. qPCR data were normalized to IgG control.
Source data are available for this figure.

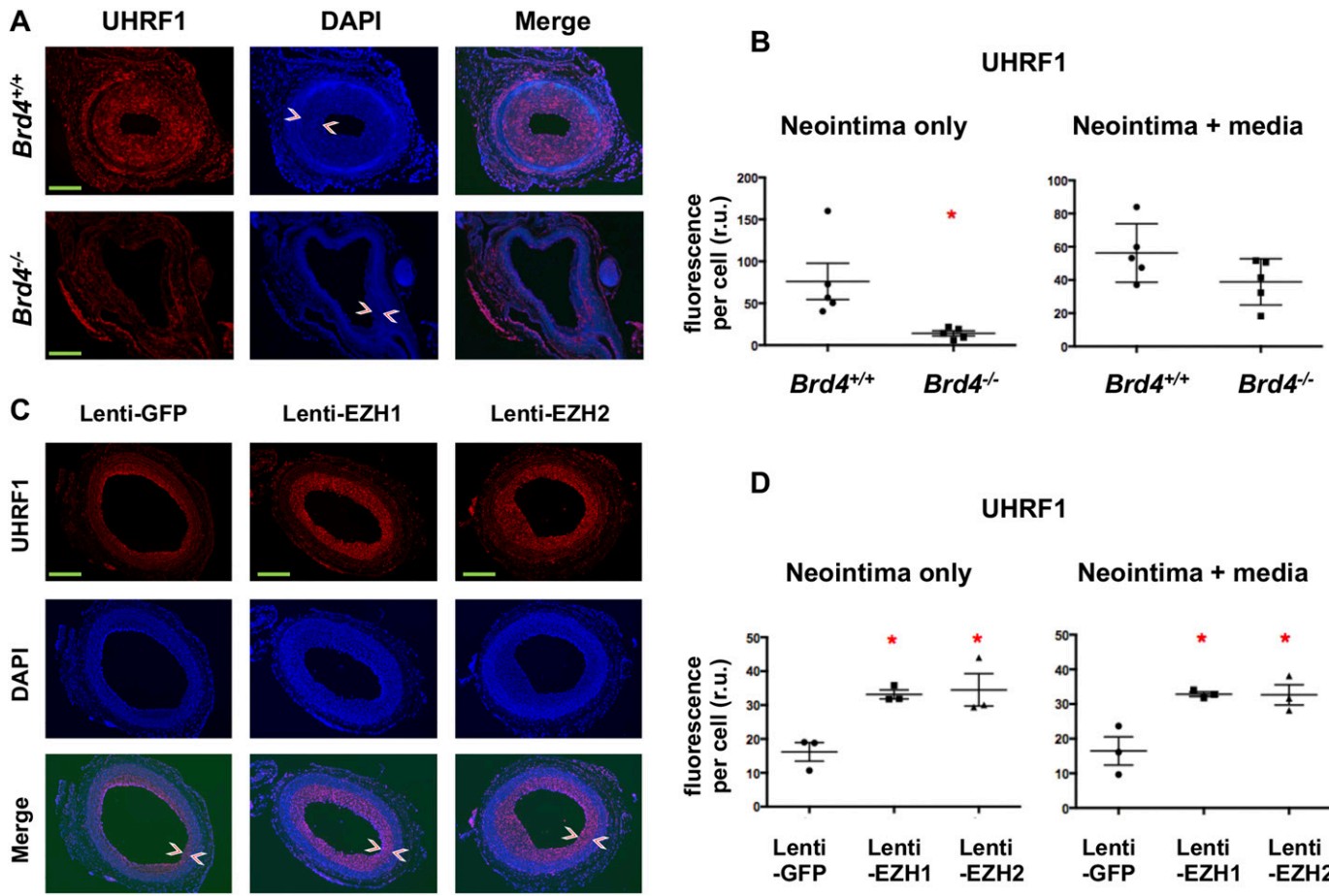

**Figure 10. Immunofluorescent detection of UHRF1 on artery cross sections.**
**(A, B)** Decrease in UHRF1 due to tamoxifen-induced BRD4 KO in wire-injured mouse femoral arteries. Neointima is demarcated between arrow heads. Scale bar: 50 $\mu$m. Fluorescence intensity was normalized to cell number. Quantification: Mean ± SEM; n = 5 mice. Statistics: nonparametric Mann–Whitney test following Shapiro–Wilk normality determination, *P < 0.05; r.u., relative unit. **(C, D)** Increase of UHRF1 after EZH1 or EZH2 overexpression in angioplasty-injured rat carotid arteries. Neointima is demarcated between arrow heads. Scale bar: 50 $\mu$m. Fluorescence intensity was normalized to cell number. Quantification: Mean ± SEM; n = 3 rats. Statistics: nonparametric Mann–Whitney test following Shapiro–Wilk normality determination, *P < 0.05 compared with GFP control; r.u., relative unit.

common carotid artery was partially dissected but not balloon-injured, and hence served as the sham control. Briefly, an incision was made in the neck of anesthetized animal. Through an opening on the left external carotid artery, the balloon was inserted and advanced ~1.5 cm into the common carotid artery, inflated (at 1.5 atm), withdrawn to the bifurcation, and then deflated before next insertion. This procedure was repeated three times. Blood flow was resumed in the common and internal carotid arteries (after ligating the external artery). The animal was maintained in general anesthesia with inhalation of 2–2.5% of isoflurane. Analgesics including carprofen and bupivacaine were injected to the animal recovering from anesthesia. Animals were euthanized in a chamber slowly filled with $CO_2$.

### Artery tissue ChIP sequencing and data processing

Tissue collection was performed at 7 d after balloon angioplasty. From the same animal, we collected the endothelially damaged central segment of the balloon-injured common carotid artery (denoted as "injured" throughout this study) and the contralateral common carotid artery without angioplasty (sham control, denoted as "uninjured"). To preserve the artery "real-time" epigenetic information, the collected tissue samples were immediately snap-frozen in liquid $N_2$. Artery tissues (injured or uninjured) from 50 rats were pooled for ChIPseq analysis at Active Motif per company standard procedures, and the sequencing raw data satisfactorily met quality control. In brief, chromatin was isolated after adding lysis buffer, followed by disruption with a Dounce homogenizer. Genomic DNA was sheared to an average length of 300–500 bp by sonicating the lysates, and the segments of interest were immunoprecipitated using an antibody (4 $\mu$g) against BRD4, H3K27ac, H3K27me3, or H3K4me1. The protein/DNA complexes eluted from beads were treated with RNase and proteinase K, crosslink was reversed, and the ChIP DNA was then purified for use in the preparation of Illumina sequencing libraries. Standard steps included end-polishing, dA-addition, adaptor ligation, and PCR amplification. The DNA libraries were quantified and sequenced on Illumina's NextSeq 500, as previously described (Ozer et al, 2018).

Sequence reads were aligned to the reference genome Rno5, peak locations were identified using Macs2 algorithm (Zhang et al, 2008) and annotated based on UCSC RefSeq. Differential peak locations were called using SICER (Zang et al, 2009). In-house shell and R scripts (https://www.r-project.org) were used for data integration. To summarize and cluster genome-wide TSS coverage as heat maps, deepTools (Ramirez et al, 2014) compute matrix and plotheatmap functions were used. IGV (http://www.broadinstitute.org/igv/) was used for visualization. Annotation files were downloaded from UCSC. To quantify ChIPseq peak values (Table S1), bigwig was converted to bedgraph and mean coverage was calculated using bedtools map for genomic regions mapped in IGV plots. Data are available through GEO with accession number GSE194390.

## Conditional KO of BRD4 and mouse femoral artery wire injury

The $Brd4^{fl/fl}$ mouse line (Dey et al, 2019) with loxP sites flanking $Brd4$ exon 3 were kindly provided by Dr. Keiko Ozato from National Institute of Child Health and Human Development (NICHD). The smooth muscle lineage-specific, tamoxifen-inducible Cre strain ($Myh11$-$CreER^{T2}$) was purchased from The Jackson Laboratory. These two strains were crossed, and the offsprings carrying $Brd4^{fl/fl}$ and/or $Myh11$-$CreER^{T2}$ were selected through genotyping as previously described (Dey et al, 2019). Genotyping PCR primers are provided in Table S2. Mice were fed with tamoxifen-citrate chow (TD.130860) for 10 d, and then with normal diet for another 7 d before femoral artery wire injury to induce IH.

Mouse femoral artery wire injury was performed as described in detail in our publication dedicated to this model (Takayama et al, 2015). Briefly, a midline incision was made in the ventral left thigh to dissect the common femoral artery. The distal and proximal ends of the femoral artery were temporally looped. An arteriotomy was made on the deep femoral artery muscular branch, through which a 0.015″ guide wire (REF#C-SF-15-15; Cook Medical) was inserted and kept stationary for 1 min. After removal of the wire, the muscular branch was ligated and blood flow was resumed. At 28 d after injury, femoral arteries were collected after perfusion fixation (with PBS first and then 4% paraformaldehyde) at a physiological pressure of 100 mm Hg. The animal was kept anesthetized with inhalation of 2.5% of isoflurane throughout the terminal procedure. Animals were euthanized in a chamber slowly filled with $CO_2$.

## Lentiviral vector construction for EZH1 or EZH2 silencing or overexpression

To construct a lentiviral vector for the expression of EZH1- or EZH2-specific shRNAs, the pLKO.1-puro empty vector was purchased from Addgene. A scrambled shRNA control and shRNAs specific for the mouse EZH1 and EZH2 genes were designed by RNAi Central (http://cancan.cshl.edu/RNAi_central/step2.cgi). The corresponding shRNA-expressing lentivectors were constructed by using the pLKO.1-puro vector as a template. For each gene, shRNAs of three different sequences were used in combination (5:3:2). The sequences that proved to be efficient are listed in Table S3. For EZH overexpression, EZH1 and EZH2 cDNA clones were purchased from Origene (Cat. no. RC202367 and Cat. no. RC202054). Lenti-EZH1-GFP and Lenti-EZH2-GFP were constructed based on these cDNA clones using a GFP-expressing lenti-vector as we previously described (Zhang et al, 2019). Lentiviruses were packaged in Lenti-X 293T cells (Cat. no. 632180; Clontech) using a three-plasmid expression system (pLKO.1-shRNAs-puro, psPAX2 and pMD2.G) as described in our recent reports (Wang et al, 2015; Zhang et al, 2019).

## Gene-editing and non-gene editing CRISPR approaches

For enhancer deletion, we first took a gene-editing approach used in our studies (El Refaey et al, 2017; Yang et al, 2019) and an online software (http://crispor.tefor.net/) for sgRNA design and off-target screening. Each of the pair of sgRNA oligos with sequences flanking the enhancer region was cloned into the lentiCRISPR v2 vector, which contains the gene of *Streptococcus pyogenes* CRISPR-Cas9 (ID52961; Addgene). The lentiCRISPR v2 vector without a sgRNA sequence was used as "scrambled" control. The sequences of the pair of sgRNAs are 5′-AGACTGGCCAGGCACTCGCGCGG-3′ (+1,161 bp from TSS) and 5′-AAATCTCTAGGGGTTGGTTGTGG-3′ (+218 bp from TSS). We also used a deactivated or dead Cas9 approach (without genome editing). The CRISPR/dCas9 transcriptional repression system was used (Kearns et al, 2014; Yang et al, 2019). The same sgRNA pair as mentioned above were subcloned into the pLV-EGFP:T2A:Neo-U6-sgRNA plasmid (ID VB210727; Vector Builder). Lentivirus was packaged as described above and used to transduce MOVAS cells.

## Intraluminal infusion of lentivirus and perivascular inhibitor drug delivery

To express a transgene or shRNA, lentivirus was infused into the balloon-injured artery wall as we recently described in detail. Briefly, immediately after angioplasty, a cannula was inserted through the external carotid artery arteriotomy, advanced past the bifurcation, and ligated to generate a sealed intraluminal space in the common carotid artery. A syringe containing lentivirus was connected to the cannula. The virus (total 150 $\mu l$, >1 × 10⁹ IFU/ml) was slowly injected, incubated for 25 min in the lumen. The lumen was then flushed repeatedly with saline containing 20 U/ml heparin and blood flow resumed. Heparin was also administered perioperatively to prevent thrombosis.

For pharmacological local treatment of injured rat carotid arteries, a thermosensitive hydrogel (AK12; Akina Inc.) was used for perivascular administration of the EZH1/2 inhibitor UNC1999, following our published method. Briefly, immediately after angioplasty, UNC1999 (10 mg/rat) or an equal amount of DMSO (vehicle control) dispersed in 400 $\mu$l AK12 gel was applied around the balloon-injured artery. The surgery was then finished as described above for the angioplasty model.

## Morphometric analysis of IH and restenosis

Paraffin cross sections (5-$\mu$m thick) were cut using a microtome (Leica) at equally spaced intervals and then stained (hematoxylin and eosin, H&E) for morphometric analysis, as described in our previous reports. Morphometric parameters as follows were measured on the sections and calculated by using ImageJ software:

area inside external elastic lamina (EEL area), area inside internal elastic lamina (IEL area), lumen area, intima area (= IEL area – lumen area), and media area (= EEL area – IEL area). Intimal hyperplasia (IH) was quantified as a ratio of intima area versus media area (I/M). Measurements were performed by an independent researcher blinded to the experimental conditions using three to six sections from each of rat. The data from all sections were pooled to generate the mean for each animal. The means from all the animals in each treatment group were then averaged, and the SEM was calculated.

## Immunofluorescence and microscopy

We used the same method as described in our recent report (Zhang et al, 2018). Briefly, artery paraffin sections were de-paraffinized and subjected to antigen retrieval. Following blocking, a primary antibody was added and incubated overnight. The sections were rinsed and incubated in a fluorescence-labeled secondary antibody for an hour. Detection of immunofluorescence was performed under an EVOS M7000 microscope (Thermo Fisher Scientific). For quantification, 3–5 immunostained sections from each animal were used. Nuclei were stained with 4′,6′-diamidino-2-phenylindole (DAPI) for counting cell numbers. Fluorescence intensity in each image field was quantified by using ImageJ software (National Institutes of Health) and normalized to cell number. The values from all sections of each animal were pooled to generate an average value. The averaged values from all the animals in each treatment group were averaged again to produce mean ± SEM.

## Immunoblotting

Cells or rat carotid artery homogenates (pulverized in liquid nitrogen) were lysed in radio-immunoprecipitation assay (RIPA) buffer containing protease inhibitors (50 mM Tris, 150 mM NaCl, 1% Nonidet P-40, 0.1% sodium dodecyl sulfate, and 10 $\mu$g/ml aprotinin). Approximately 15–30 $\mu$g of proteins from each sample were separated via sodium dodecyl sulfate-polyacrylamide gel electrophoresis on a 10% gel. The proteins were then transferred to a polyvinylidene difluoride membrane and detected by immunoblotting. The antibody sources and dilution ratios are listed in Table S4. Specific protein bands on the blots were illuminated by applying enhanced chemiluminescence reagents (Cat. no. 32106; Thermo Fisher Scientific) and then recorded with an Azur LAS-4000 Mini Imager (GE Healthcare Bio-Sciences). Band intensity was quantified by using ImageJ software.

## Assays for proliferation and migration

Proliferation was determined by using the CellTiter-Glo Luminescent Cell Viability kit (Promega) following the manufacturer's instructions. Wild-type or lentiviral-infected MOVAS (a mouse vascular smooth muscle line) cells were seeded in 96-well plates at a density of 2,000 cells per well with a final volume of 200 $\mu$l DMEM (10% FBS). Cells were starved with 0.5% FBS overnight and then stimulated with PDGF-BB (20 ng/ml). At 72 h of PDGF-BB treatment, plates were decanted, refilled with 50 $\mu$l CellTiter-Glo reagent/50 $\mu$l phosphate-buffered saline per well, and incubated at room temperature for 10 min before reading in a FlexStation 3 Benchtop Multi-Mode Microplate Reader (Molecular Devices).

To determine cell migration, scratch (wound healing) assay was performed as described in our previous report (Wang et al, 2015). Briefly, wild-type or lentiviral-infected MOVAS cells were cultured to a 90% confluency in six-well plates and then starved overnight. A sterile pipette tip was used to generate an ~1 mm cell-free gap. Dislodged cells were washed away with PBS. Plates were then refilled with fresh medium containing 20 ng/ml of PDGF-BB and incubated for 24 h. Calcein AM was then added (2 $\mu$M) to illuminate the cells. After a 15-min incubation, cells were washed three times with PBS, and images were then taken. Cell migration was quantified by ImageJ software based on the change in the width of the cell-free gap before and after PDGF-BB stimulation.

## Quantitative real-time polymerase chain reaction (qPCR)

Assays were performed following our published methods. Briefly, total ribonucleic acid was isolated from cultured cells or rat carotid arteries (pulverized in liquid nitrogen) by using a Trizol reagent (Thermo Fisher Scientific) following the manufacturer's protocol. Potential contaminating genomic deoxyribonucleic acid (DNA) was removed by using gDNA Eliminator columns provided in the kit. Total ribonucleic acid of 1 $\mu$g was used for the first-strand complementary DNA synthesis (Thermo Fisher Scientific). Quantitative real-time polymerase chain reaction was performed by using Quant Studio 3 (Thermo Fisher Scientific). Each complementary DNA template was amplified in triplicate PerfeCTa SYBR Green SuperMix (Quantabio). Primers are listed in Table S5.

## Chromatin immunoprecipitation (ChIP)-qPCR

ChIP was performed as described in detail in our recent report (Xie et al, 2020). The Pierce Magnetic ChIP kit (26157; Thermo Fisher Scientific) was used. In brief, MOVAS cells treated with solvent control or 20 ng/ml of PDGF-BB were subjected to crosslinking. The washed cells were lysed for nuclei extraction. After DNA digestion with MNase, the nuclei were recovered and burst by sonication. ChIP was then performed by incubating chromatin extracts with a H3K27ac antibody (or IgG control included in the kit) and ChIP-grade Protein A/G Magnetic Beads. Proteins and RNAs were digested and DNA fragments were purified, which were used for qRT-PCR. The primers are listed in Table S6.

## Statistical analysis

Data are presented as mean ± SEM. In statistical analysis, one-way ANOVA followed by post-hoc Bonferroni test was applied to multigroup comparison. For two-group comparison, either parametric $t$ test or nonparametric Mann–Whitney test was applied, on the basis of data normality determination using Shapiro–Wilk normality test, as specified in figure legends. Statistical significance was set at $P <$ 0.05 using GraphPad Prism (Graphpad Software) except for ChIPseq data. For ChIPseq data specifically, statistical analyses were performed using SAS/STAT software, version 9.2 (SAS Institute, Inc.) (for SICER-df_intervals, see Table S7).

# Data Availability

The data that support the findings of this study are available through GEO with accession number GSE194390 or from the corresponding authors upon reasonable request.

# Supplementary Information

# Acknowledgements

We thank Dr. Keiko Ozato (Section on Molecular Genetics of Immunity, National Institute of Child Health and Human Development) for kindly providing the *Brd4*$^{fl/fl}$ mouse strain. We also thank Drs. Matthew Stratton and Noah Weisleder for discussions This work was supported by NIH grants R01 HL133665 (to L-W Guo), R01HL-143469 and R01HL-129785 (to KC Kent, L-W Guo), AHA pre-doctoral award 17PRE33670865 (to M Zhang), AHA post-doctoral award 20POST35210967(to M Zhang) and Overseas Research Fellowships and The Uehara Memorial Foundation in Japan (to T Shirasu). R Han is supported by NIH grants (R01HL116546, R01HL159900, R01AR070752) and a Parent Project Muscular Dystrophy award.

## Author Contributions

M Zhang: data curation and investigation.
G Urabe: data curation and formal analysis.
HG Ozer: formal analysis, investigation, and visualization.
X Xie: data curation and methodology.
A Webb: formal analysis, investigation, and visualization.
T Shirasu: data curation.
J Li: data curation.
R Han: formal analysis, supervision, funding acquisition, investigation, methodology, and gene editing methodology.
C Kent: supervision and funding acquisition.
B Wang: conceptualization, formal analysis, investigation, and writing—review and editing.
L-W Guo: conceptualization, supervision, funding acquisition, project administration, and writing—original draft, review, and editing.

## Conflict of Interest Statement

The authors declare that they have no conflict of interest.

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
