## [Reviewer comments · Life Science Alliance]

Life Science Alliance

Angioplasty induces epigenomic remodeling in injured arteries

Mengxue Zhang, Go Urabe, Hatice Gulcin Ozer, Xiujie Xie, Amy Webb, Takuro Shirasu, jing Li, Renzhi Han, Craig Kent, Bowen Wang, and Lian-Wang Guo

DOI: <https://doi.org/10.26508/lsa.202101114>

Corresponding author(s): Lian-Wang Guo, University of Virginia and Bowen Wang, UVA

Review Timeline:

Submission Date:	2021-05-05
Editorial Decision:	2021-07-09
Revision Received:	2021-12-09
Editorial Decision:	2022-01-04
Revision Received:	2022-01-25
Accepted:	2022-01-26

Scientific Editor: Novella Guidi

Transaction Report:

July 9, 2021

Re: Life Science Alliance manuscript #LSA-2021-01114

Dr. Lian-Wang Guo
University of Virginia
409 Lane Rd, MR4(building), Room 2146
Charlottesville, VA 22908

Dear Dr. Guo,

Thank you for submitting your manuscript entitled "Angioplasty induces epigenomic remodeling in injured arteries" to Life Science Alliance. The manuscript was assessed by expert reviewers, whose comments are appended to this letter. As you will note from the reviewers' comments below, all reviewers are quite positive and excited about the work that in their views offers new insight on potential pharmacologic targets to stop the progression of intimal hyperplasia. However, they do raise some important concerns that need to be addressed in the revised version. Besides Reviewer 1 point 9, which we considered not necessary for the scope of this manuscript, please address all the other concerns raised by the reviewers. We, thus, encourage you to submit a revised version of the manuscript back to LSA that responds to all of the reviewers' points.

Thank you for this interesting contribution to Life Science Alliance. We are looking forward to receiving your revised manuscript.

Sincerely,

B. MANUSCRIPT ORGANIZATION AND FORMATTING:

Reviewer #1 (Comments to the Authors (Required)):

In their manuscript entitled "Angioplasty induces epigenomic remodeling in injured arteries", Zhang and co-workers describe the loci-specific H3K27Me3 redistribution that associates with the development and progression of intimal hyperplasia. Given that the epigenomic enzymes that regulate H3K27Me3 and H3K27Ac are both known pharmacologic targets in the oncology field, the data reported offers new insight on potential pharmacologic targets to stop the progression of intimal hyperplasia. Some critiques to the manuscript, however, must be addressed.

Major comments:

1. How do the authors correct for the loss of endothelial cells in their experiment. Did the authors use balloon-denuded arteries to ensure these resemble the uninjured samples? If not, how can the authors be sure that the observed differences do not reflect the presence of an endothelial cell layer?
2. Page 4. ". Gene annotation showed that gene-regulatory factors are top scored in both Cluster-1 and Cluster-2." The full data of such analysis need to be presented in the manuscript (or supplement) to substantiate these claims. Also, it would be of interest to show the overlap between BRD4 ChIP and H3K27Ac ChIP in the data supplement to evidence their co-regulation. Although the authors note in the results that BRD4 peaks show a 93% overlap with H3K27Ac, this does not become apparent from fig. 1C that shows drastic differences between the BRD4 and H3K27Ac ChIP.
3. Figure 1B. Bean plots need statistical evaluation and p-values need to be inserted in the figure to ensure that claimed differences are present.
4. Figure 2A should be quantified to ensure enrichment of BRD4 and H3K27Ac peaks at EZH2 and Nrp2 in replicate experiments.
5. Figure 3 should be extended to include EZH2 itself. Although the authors state to measure "H3K27me3 on artery cross sections to read out EZH2's function as the writer (Figure 3, E and F)", this conclusion cannot be drawn from the presented data because multiple factors besides EZH2 influence the presence of H3K27Me3. Such factors would include the expression and activity of EZH2 (that maintains H3K27Me3), UTX and JMJD3 (which demethylate H3K27). In the absence of data on EZH2, the authors can only conclude that the genetic loss of BRD4 in SMC results in the accumulation of H3K27Me3. This is irrespective of the in vitro results presented in figs 4A-B, as these not necessarily reflect what occurs in vivo. Also, in the absence of an intervention at the level of BRD4, also fig. 4C does not provide proof for the direct regulation of EZH2 expression by BRD4 in intimal hyperplasia.
6. Figures 5E and H should be quantified.
7. Figure 6A should be quantified to ensure enrichment of BRD4 and H3K27Ac peaks at Uhrf1. Figure E and F should be extended to include BRD4 and H3K27Ac and quantified.
8. Datasets obtained in the various ChIP experiments should be made available in public repositories.
9. The translational element of this work could be greatly enhanced if the authors could show that the reciprocity between BRD4 and EZH2 is also present in human intimal hyperplasia and the authors should seriously consider to extend their key observations to human angioplasty samples.

Minor comments:

1. Representative heavily cropped immunoblots are shown in figures ..., ... and ... The authors should provide all full images of all membranes analyzed in the manuscript to allow validation of these quantifications by the readership.
2. In essence, figures 1B and 1C show similar data, albeit using different cut-off values. Fig. 1C in contrast to Fig. 1B allows to estimate how many genes were increased/decreased in the ChIP experiments. For simplicity sake, it would be preferred to maintain Fig. 1B in the main manuscript and move fig. 1C to the data supplement.

Reviewer #2 (Comments to the Authors (Required)):

This paper provides interesting and new insight in epigenetic response subsequent to in vivo vascular intima injury. By exploiting ChIPseq in angioplasty-damaged carotid arteries the authors provide evidences of an increased H3K27me3, and H3K27ac binding intensity to DNA regions near TSS. This specific enrichment moved from proliferative to anti-proliferative genes. Interestingly, BRD4 / H3K27ac and H3K27me3 peaks are mutually exclusive. Then the authors demonstrate that the loci-specific redistribution of H3K27me3 requires a regulatory cascade involving the epigenetic regulators EZH2 and BRD4. The

experiments are rigorously planned and the proposed *in vivo* approach is a step forward in the field. The results obtained contribute to explain the changes of activation state (quiescent vs activated/proliferating) of VSMC during intimal response to endothelial injury. However there are some issues that need to be solved.

CRITICISMS

Fig 2E. The effect of CRISPR technology on DNA sequencing of region with the EZH2 enhancer should be shown.

Fig 3E. Because the authors advance the hypothesis that the regulatory role of H3K27ac and H3K27me3 is mutually exclusive and they participate to the regulation of cell proliferation, it is important to correlate the expression of H3K27me3 with a mitotic marker at single cell level.

Fig 3. Which is the expression of EZH2 in BRD4 null mice?

Fig 3. To better validate the proposed sequence of molecular events BRD4 → EZH2/H3K27me3 I suggest to perform LV mediated EZH2 gain and loss of function approach shown in figure 4 in artery injury performed in BRD4 null mice

Fig 5. Does EZH2 overexpression positively modulate VSMC migration and proliferation?

The experiments in Figure 6 are interesting because they indicate a new target (UHRF1) of EZHw.

However, to support panel G, it is necessary to show the expression of this molecule in an *in vivo* setting both in wild-type and BRD4 null mice.

Reviewer #3 (Comments to the Authors (Required)):

Neointima hyperplasia (IH) represents the major cause of restenosis upon angioplasty and is caused by proliferation and migration of vascular smooth muscle cells (SMC). Research focuses on the identification of novel mechanisms to interfere with SMC activation in order to inhibit IH. Even though studies have shown that inhibition of epigenetic modifiers regulate IH, genome-wide studies of certain histone-marks remained to be elusive.

In the manuscript "Angioplasty induces epigenomic remodeling in injured arteries" Zhang and coworkers address this gap of knowledge by generating epigenetic profiles of rat carotid arteries after injury vs. control. These data revealed an induction and re-distribution of the repressive histone mark H3K27me3 affecting the expression of IH-related genes. Further data show that the mediator of H3K27 trimethylation, Ezh2 is induced in a BRD4-dependent manner *in vitro* and SMC-specific deletion of BRD4 impairs IH *in vivo*. Similarly, Ezh1 or Ezh2 regulates IH and SMC activation which is investigated *in vivo* and *in vitro* by pharmacological inhibition or selective lentiviral-based gene silencing or overexpression. Altogether, these data enhance our understanding of the epigenetic mechanisms involved in IH.

In general, this manuscript is of interest in the field of translational stenotic research as it not only provides a comprehensive resource tool of genome-wide histone modifications but also demonstrates solid *in vivo* data.

However, the manuscript would benefit from comprehensive editing in order to strengthen the findings of the paper and further stress the central theme. The authors build the manuscript on the novel ChIPseq dataset. However, these data are presented as n=1 and therefore the authors should aim for validation ChIP experiments. In addition, thorough ChIPseq data analysis is missing (see comments below). Similarly, the correlation of the epigenetic findings with corresponding gene expression data (selfmade or from databases) would underline the functional relevance of the identified epigenetic mark redistribution. Next, the manuscript focuses on the role of Ezh2 and Ezh1 in IH and parts of the data presented here has been shown before. The authors are encouraged to stress the novel findings more and clearly state what has been described before (moving these data to the supplements). At the end of the manuscript, it is indicated that blocking Ezh2 impairs H3K37me3 re-distribution upon injury. This claim would need to be confirmed by corresponding ChIP(seq) data upon either Ezh2 deletion or pharmacological Ezh2 inhibition (side note: there are novel Ezh gene specific inhibitors available) during injury. In greater detail, this reviewer suggests to consider the following points to increase the clarity and the impact of the manuscript:

The 1st results section describes the method and the model system. This is not a finding and should therefore be included in following result section.

1. Please revise the 1st sentence in the legend of Figure S1.

The 2nd results section refers to H3K27me3 remodeling upon angioplasty which is shown by ChIPseq analysis. Figure 1A convincingly shows an increase in the given histone mark at certain TSS (cluster 1). However, the text of the related figure is misleading and the authors should take the following into account:

2. It is mentioned several times in the text that H3K27ac and BRD4 are primarily enhancer marks. Yet, (i) Figure 1A focuses on TSS +/-5kb and (ii) both, the localization of H3K27ac and BRD4 has not only been shown on enhancers but also on TSS/promoter (e.g. Zhang et al. J Biol Chem 2012; Anders et al. Nat Biotech 2013) which is basically also verified in Figure 1A. Please carefully revise your conclusions. How can "enhancer-associated BRD4 peaks" be located at promoters?

3. The genomic features associated with H3K27ac and H3K27me3 are nicely introduced but is missing for H3K4me1.
4. Figure 1B - legend: There is no transcript abundance of genes shown here. Please revise. Moreover, in the text it says that "gene annotation showed that gene-regulatory factors are top scored in both (..) ". Please include the mentioned gene annotation in the presented figure and give a list of those factors. In the same line the analysis would benefit to a great extent from a correlation of the ChIPseq data to gene expression data (own data or from public database).
5. In general, it would improve the manuscript to elaborate on the ChIPseq data in more detail:
 - not only focusing on promoter but also enhancer regions - are they regulated differently upon injury?
 - including info about the genes next to certain epigenetic marks - are certain pathways affected by epigenetic remodeling?
 - motif enrichment in regions of interest?

The 3rd section focuses on the regulation of Ezh2 expression on the epigenetic level upon angioplasty. While the injury-induced enrichment of H3K27ac on the Ezh2 promoter is clearly visible in the genome browser view (Figure 2A), it is less obvious for BRD4.

6. To support the corresponding statement in the text the authors are encouraged to include the quantification of the peak intensities.
7. Please add qPCR data showing the induction of Ezh2 and Nrp2 upon injury. The authors claim that the peak region depicted in panel A (enriched for H3K27ac and BRD4) marks an active enhancer. However, the H3K4me1 ChIPseq data does not support that. The histone modification H3K4me1 is a common identifier of enhancers regions, independent of activity status.
8. Adding ChIP data of H3K4me3 (promoter mark) would clarify the status of this region of interest.
9. In order to exclude that the used sgRNA targeting the region of interest is showing an effect independent of interfering with transcription, the reviewer recommends to add data of a sgRNA targeting an intronic region further upstream as a control.

The 4th section shows that SMC-specific deletion of Brd4 reduces IH and results in decreased H3K27me3 upon injury.

10. The deletion efficiency by TAM administration needs to be quantified.
11. In the same line, to support the link to Ezh2, please add the expression of Ezh2 upon injury in general and upon Brd4 deletion in SMC in vivo.
12. The last part of this results section refers to in vitro data that is currently presented in Figure 4. In order to keep the structure consistent, these data would be more suitable in Figure 3.

The following 5th section addresses the role of Ezh2 in IH formation by gain and loss of function experiments convincingly showing that Ezh2 promotes IH.

13. In order to verify that the effects shown in panel G-I are Ezh-dependent, the authors should verify the expression of Ezh1 and Ezh2 either by (GFP) staining or expression quantification of isolated SMCs from arteries.
14. In the legend: please explain the abbreviations N and M used in panel E.

In the 6th section the manuscript focuses on in vitro studies to underline the role of both Ezh1 and Ezh2 in SMC proliferation and migration induced by PDGFB. This reviewer encourages the authors to address the following points:

15. The picture in panel B depicting the 0h time point should be replaced by a more convincing picture (GFP staining)
16. Panel C and F: Comparing the control samples (scrambled and Lenti-GFP) the treatment with PDGFB induces K27me3 in panel C while it not induced (rather repressed) in panel F. Please comment on that.
17. In the same line, the control treatments in Panel E and H should be same/similar in migration/ scratch closure - but they are remarkable different. Please comment on that as well.

Section #7 (Upper Part of Figure 6) claims Uhrf1 as a novel target of Ezh isoforms. This finding is linked to the first figure and should rather be presented as a supplement of it. Moreover, the link has been published previously.

The last section (Fig. 6 E-G and Fig. 7) addresses the redistribution of H3K27me3 upon angioplasty. A point that has been raised in the first figure. Therefore, the authors should consider to shift these analyses to the first part of the manuscript and demonstrate the re-distribution on a genomic scale of H3K27ac/me3 rather on single gene expression level.

Minor comments:

Figure legends:

- technical details in legends should be removed and described in the methods section only
- partially missing n numbers (qPCR data, WB data)

Results section:

- The concluding remarks after each results section is partially missing.
- Some sentences set the results in perspective with other data which is rather discussion

Methods:

Please deposit your ChIPseq data in public database (like GEO) to make it available and a valid source for further research. At least the peak regions should be provided as a supplement (e.g bed file format).

Dear Dr. Guidi,

We appreciate very much the time and input the reviewers and editors have dedicated to improving our manuscript (LSA-2021-01114), and we are glad the reviewers found our work interesting. The reviewers' comments are extremely helpful. Following their suggestions, we have performed new experiments, added new figures, and carefully revised the manuscript. Below please find our point-by-point responses.

New figures include:

Figure 1B, Figure 4D, Figure 5C, Figure 6A/B, Figure 7B, Figure 8D and H, Figure 9J, Figure 10, and Figure S1, Figure S2, Figure S4, Figure S5, Figure S6A, Figure S7, Figure S9.

July 9, 2021

Re: Life Science Alliance manuscript #LSA-2021-01114

Dr. Lian-Wang Guo
University of Virginia
409 Lane Rd, MR4(building), Room 2146
Charlottesville, VA 22908

Dear Dr. Guo,

Thank you for submitting your manuscript entitled "Angioplasty induces epigenomic remodeling in injured arteries" to Life Science Alliance. The manuscript was assessed by expert reviewers, whose comments are appended to this letter. As you will note from the reviewers' comments below, all reviewers are quite positive and excited about the work that in their views offers new insight on potential pharmacologic targets to stop the progression of intimal hyperplasia. However, they do raise some important concerns that need to be addressed in the revised version. Besides Reviewer 1 point 9, which we considered not necessary for the scope of this manuscript, please address all the other concerns raised by the reviewers. We, thus, encourage you to submit a revised version of the manuscript back to LSA that responds to all of the reviewers' points.

You will be guided to complete the submission of your revised manuscript and to

fill in all necessary information. Please get in touch in case you do not know or remember your login name.

Thank you for this interesting contribution to Life Science Alliance. We are looking forward to receiving your revised manuscript.

Sincerely,

Reviewer #1 (Comments to the Authors (Required)):

In their manuscript entitled "Angioplasty induces epigenomic remodeling in injured arteries", Zhang and co-workers describe the loci-specific H3K27Me3 redistribution that associates with the development and progression of intimal hyperplasia. Given that the epigenomic enzymes that regulate H3K27Me3 and H3K27Ac are both known pharmacologic targets in the oncology field, the data reported offers new insight on potential pharmacologic targets to stop the progression of intimal hyperplasia. Some critiques to the manuscript, however, must be addressed.

>> We appreciate very much the reviewer's dedication to reviewing our manuscript.

Major comments:

1. How do the authors correct for the loss of endothelial cells in their experiment. Did the authors use balloon-denuded arteries to ensure these resemble the uninjured samples? If not, how can the authors be sure that the observed differences do not reflect the presence of an endothelial cell layer?

>> This is an excellent question. Realistically there has been no angioplasty model available that can maintain equal numbers of endothelial cells (ECs) in injured and uninjured arteries. Even if there were, the injured and uninjured ECs would be phenotypically different. The rat artery angioplasty model is by far the best for our purpose of epigenomic study on intimal hyperplasia, for the following reasons: 1. Smooth muscle cells (SMCs) comprise the major cell population in the artery wall, in contrast to a much smaller number of ECs that only form a single layer. 2. Angioplasty damages the EC lining but does not completely remove ECs. 3. We detected injury-induced upregulation of neointima markers UHRF1 and NRP2 which were previously reported to be upregulated mainly in SMCs in vivo. 4. Our ChIPseq result of injury-induced increase of H3K27ac binding at *Ezh2* and *Uhrf1* was validated in vitro in SMCs by ChIP-qPCR (Figure 9J). Therefore, we thought the in vivo ChIPseq results could provide interesting leads for further research.

Please see the revision in Discussion highlighting this study limitation, the bottom of Page 9:

"However, with future translation in mind, it is important to note limitations in this study. These include the use of healthy animals without human-like disease backgrounds, uncertain contribution of ECs in the ChIPseq samples, and the lack of data from human subjects or samples."

2. Page 4. ". Gene annotation showed that gene-regulatory factors are top scored in both Cluster-1 and Cluster-2." The full data of such analysis need to be presented in the manuscript (or supplement) to substantiate these claims.

>> Thanks. We now add the gene ontology analysis in Figure 1B.

Also, it would be of interest to show the overlap between BRD4 ChIP and H3K27Ac ChIP in the data supplement to evidence their co-regulation. Although the authors note in the results that BRD4 peaks show a 93% overlap with H3K27Ac, this does not become apparent from fig. 1C that shows drastic differences between the BRD4 and H3K27Ac ChIP.

>> Venn diagrams showing BRD4 and H3K27ac peak overlap are now presented, please see Figure S3, and revision in Page 4, Paragraph 3:

"The majority of the BRD4 ChIPseq peaks overlapped with that of H3K27ac (see Venn diagrams in Figure S3), as expected since both are associated with active enhancers".

3. Figure 1B. Bean plots need statistical evaluation and p-values need to be inserted in the figure to ensure that claimed differences are present.

>> Great suggestion. Please see P values added to new Figure 2A.

4. Figure 2A should be quantified to ensure enrichment of BRD4 and H3K27Ac peaks at EZH2 and Nrp2 in replicate experiments.

>> This is a nice idea. We now add Table S1 to include ChIPseq peak coverage values.

5. Figure 3 should be extended to include EZH2 itself. Although the authors state to measure "H3K27me3 on artery cross sections to read out EZH2's function as the writer (Figure 3, E and F)", this conclusion cannot be drawn from the presented data because multiple factors besides EZH2 influence the presence of H3K27Me3. Such factors would include the expression and activity of EZH2 (that maintains H3K27Me3), UTX and JMJD3 (which demethylate H3K27). In the absence of data on EZH2, the authors can only conclude that the genetic loss of BRD4 in SMC results in the accumulation of H3K27Me3. This is irrespective of the in vitro results presented in figs 4A-B, as these not necessarily reflect what occurs in vivo. Also, in the absence of an intervention at the level of BRD4, also fig. 4C does not provide proof for the direct regulation of EZH2 expression by BRD4 in intimal hyperplasia.

>> We fully agree. We now add EZH2 immunostaining data in Figure 6A/B, which provides direct evidence of reduced EZH2 expression due to BRD4 KO. Please also see the corresponding revision in Page 6, Paragraph 2:

"We then measured on artery cross-sections the levels of EZH2 (Figure 6, A and B) and its catalytic product H3K27me3 (Figure 6, C and D). Immunofluorescence indicated that BRD4 knockout substantially reduced EZH2 in the neointimal layer;"

6. Figures 5E and H should be quantified.

>> Done. Please see new Figure 8, C/D and G/H, and revision in Page 6, Paragraph 4:

"Pre-treatment with the pan-EZH1/2 inhibitor UNC1999 concentration-dependently inhibited PDGF-induced SMC proliferation and migration (Figure S8). Furthermore, in an isoform-specific manner, silencing either EZH2 or EZH1 with shRNA markedly inhibited PDGF-induced SMC proliferation and migration (Figure 8, A-D)."

7. Figure 6A should be quantified to ensure enrichment of BRD4 and H3K27Ac peaks at Uhrf1. Figure E and F should be extended to include BRD4 and H3K27Ac and quantified.

>> Accomplished. The full gene track profiles are now presented in Figure 3, and the quantified peak values are listed in Table S1. Please see revision in Page 5, Paragraph 1:

“As illustrated by Figure 3A and 3B (ChIPseq peak coverage in Table S1), H3K27me3 ChIPseq peaks intensified at *P57* but ebbed at *Ccnd1* after arterial injury; in contrast, H3K27ac occupancy markedly increased instead at *Ccnd1*.”

8. Datasets obtained in the various ChIP experiments should be made available in public repositories.

>> Indeed. This is being done per journal policy.

9. The translational element of this work could be greatly enhanced if the authors could show that the reciprocity between BRD4 and EZH2 is also present in human intimal hyperplasia and the authors should seriously consider to extend their key observations to human angioplasty samples.

>> Please see Editor Dr. Novella Guidi's specific comment on this point in the decision letter attached above.

Taking the reviewer's nice suggestion, we now highlight this study limitation in Discussion, please see the bottom of Page 9:

“However, with future translation in mind, it is important to note limitations in this study. These include the use of healthy animals without human-like disease backgrounds, uncertain contribution of ECs in the ChIPseq samples, and the lack of data from human subjects or samples.”

Minor comments:

1. Representative heavily cropped immunoblots are shown in figures ..., ... and ... The authors should provide all full images of all membranes analyzed in the manuscript to allow validation of these quantifications by the readership.

>> We fully agree. The original blots are now included in a supplemental file.

2. In essence, figures 1B and 1C show similar data, albeit using different cut-off values. Fig. 1C in contrast to Fig. 1B allows to estimate how many genes were increased/decreased in the ChIP experiments. For simplicity sake, it would be preferred to maintain Fig. 1B in the main manuscript and move fig. 1C to the data supplement.

>> The reviewer is correct. Since we have rearranged figures as needed for the revision, we find it is easier to describe the results if having the previous Figure 1B and 1C together (now in Figure 2).

Reviewer #2 (Comments to the Authors (Required)):

This paper provides interesting and new insight in epigenetic response subsequent to in vivo vascular intima injury. By exploiting ChIPseq in angioplasty-damaged carotid arteries the authors provide evidences of an increased H3K27me3, and H3K27ac binding intensity to DNA regions near TSS. This specific enrichment moved from proliferative to anti-proliferative genes. Interestingly, BRD4 / H3K27ac and H3K27me3 peaks are mutually exclusive. Then the authors demonstrate that the loci-specific redistribution of H3K27me3 requires a regulatory cascade involving the epigenetic regulators EZH2 and BRD4. The experiments are rigorously planned and the proposed in vivo approach is a step forward in the field . The results obtained contribute to explain the changes of activation state (quiescent vs activated/proliferating) of VSMC during intimal response to endothelial injury. However there some issues that need to be solved.

>> Thank you for your comments.

CRITICISMS

Fig 2E. The effect of CRISPR technology on DNA sequencing of region with the EZH2 enhancer should be shown.

>> We understood the reviewer's concern over possible off-target effects of the CRISPR technology. We therefore performed a new experiment using a secondary method designed to interrupt *Ezh2* enhancer's function without genome editing, and we observed reduced EZH2 expression. Please see Figure 4D, and revision in Page 5, the middle of Paragraph 2:

"Since genome editing possibly involves off-targets, we then applied a non-genome-editing method (used in our recent report) which is gaining popularity for minimizing off-target concerns. In principle, guided by sgRNA, deactivated (or dead) Cas9 fused with a repressor protein binds to the targeted enhancer (yet without cutting) thereby hindering its transcription-enhancing function. Using this approach, we again observed reduced EZH2 mRNA (sgRNA vs scrambled, Figure 4D)."

Fig 3E. Because the authors advance the hypothesis that the regulatory role of H3K27ac and H3K27me3 is mutually exclusive and they participate to the regulation of cell proliferation, is is important to correlate the expression of H3K27me3 with a mitotic marker at single cell level.

>> Yes. We now show that immunostained mitotic marker PCNA overlaps with H3K27me3 in individual nuclei in the neointima. Please see Figure S7, and revision in Page 6, the middle of Paragraph 3:

"Consistently, immunostained mitotic marker proliferating cell nuclear antigen (PCNA) overlapped with H3K27me3 in the nuclei of periluminal neointimal cells (Figure S7)."

Fig 3. Which is the expression of EZH2 in BRD4 null mice?

>> We now add EZH2 immunostaining data in Figure 6A/B, please see revision in Page 6, Paragraph 2:

“We then measured on artery cross-sections the levels of EZH2 (Figure 6, A and B) and its catalytic product H3K27me3 (Figure 6, C and D). Immunofluorescence indicated that BRD4 knockout substantially reduced EZH2 in the neointimal layer;”

Fig 3. To better validate the proposed sequence of molecular events BRD4 -> EZH2/H3K27me3 I suggest to perform LV mediated EZH2 gain and loss of function approach shown in figure 4 in artery injury performed in BRD4 null mice

>> To validate the proposed BRD4 -> EZH2 sequence, we now provide immunostaining data of reduced EZH2 expression as a result of BRD4 KO (Figure 6A/B). To further provide in vivo evidence for the EZH2-> H3K27me3 sequence, we now show increased H3K27me3 in arteries as a result of EZH2 gain of function (Figure 7B). Please see revision in Page 6, Paragraph 3:

“We found that compared to the GFP control, increasing EZH2 heightened H3K27me3 (Figure 7B)”.

Fig 5. Does EZH2 overexpression positively modulate VSMC migration and proliferation?

>> Yes. Please see the quantified data in new Figure 8, and revision at the top of Page 7:

“In further support of this conclusion, lentivirus-mediated gain-of-function experiments indicated that increasing either EZH2 or EZH1 enhanced SMC proliferation and migration (Figure 8, E-H).”

The experiments in Figure 6 are interesting because they indicate a new target (UHRF1) of EZHw. However, to support panel G, it is necessary to show the expression of this molecule in an in vivo setting both in wild-type and BRD4 null mice.

>> Accomplished. Please see 10A, and revision in Page 7, the bottom of Paragraph 3:

“Indeed, while UHRF1 markedly decreased due to BRD4 knockout in mouse arteries (Figure 10, A and B), it was increased by EZH2 (and EZH1) gain-of-function in injured rat arteries (Figure 10, C and D).”

Reviewer #3 (Comments to the Authors (Required)):

Neointima hyperplasia (IH) represents the major cause of restenosis upon angioplasty and is caused by proliferation and migration of vascular smooth

muscle cells (SMC). Research focuses on the identification of novel mechanisms to interfere with SMC activation in order to inhibit IH. Even though studies have shown that inhibition of epigenetic modifiers regulate IH, genome-wide studies of certain histone-marks remained to be elusive.

In the manuscript "Angioplasty induces epigenomic remodeling in injured arteries" Zhang and coworkers address this gap of knowledge by generating epigenetic profiles of rat carotid arteries after injury vs. control. These data revealed an induction and re-distribution of the repressive histone mark H3K27me3 affecting the expression of IH-related genes. Further data show that the mediator of H3K27 trimethylation, Ezh2 is induced in a BRD4-dependent manner in vitro and SMC-specific deletion of BRD4 impairs IH in vivo. Similarly, Ezh1 or Ezh2 regulates IH and SMC activation which is investigated in vivo and in vitro by pharmacological inhibition or selective lentiviral-based gene silencing or overexpression. Altogether, these data enhance our understanding of the epigenetic mechanisms involved in IH.

In general, this manuscript is of interest in the field of translational stenotic research as it not only provides a comprehensive resource tool of genome-wide histone modifications but also demonstrates solid in vivo data.

>> Thank you for your comments.

However, the manuscript would benefit from comprehensive editing in order to strengthen the findings of the paper and further stress the central theme. The authors build the manuscript on the novel ChIPseq dataset. However, these data are presented as n=1 and therefore the authors should aim for validation ChIP experiments. In addition, thorough ChIPseq data analysis is missing (see comments below). Similarly, the correlation of the epigenetic findings with corresponding gene expression data (selfmade or from databases) would underline the functional relevance of the identified epigenetic mark redistribution. Next, the manuscript focuses on the role of Ezh2 and Ezh1 in IH and parts of the data presented here has been shown before. The authors are encouraged to stress the novel findings more and clearly state what has been described before (moving these data to the supplements).

>> Thank you. These are all very helpful suggestions. We have performed new experiments to address these comments, made comprehensive editing, and stressed the novel findings more clearly. Some previously described data have been moved to supplements (Figure S6B and Figure S8).

At the end of the manuscript, it is indicated that blocking Ezh2 impairs H3K37me3 re-distribution upon injury. This claim would need to be confirmed

by corresponding ChIP(seq) data upon either Ezh2 deletion or pharmacological Ezh2 inhibition (side note: there are novel Ezh gene specific inhibitors available) during injury.

>> We recognize that it was an overstatement. Please see revision in Page 10, Paragraph 1:

“On the other hand, pharmaceutical development of “epi-drugs” is rapidly advancing, mainly in the cancer field. To seize this momentum for improved treatments of vascular diseases, more research is warranted to delineate the specific roles of various chromatin modulators and relationships thereof in the IH disease background.”

In greater detail, this reviewer suggests to consider the following points to increase the clarity and the impact of the manuscript:

The 1st results section describes the method and the model system. This is not a finding and should therefore be included in following result section.

>> We agree. This has been fixed accordingly.

1. Please revise the 1st sentence in the legend of Figure S1.

>> Done.

The 2nd results section refers to H3K27me3 remodeling upon angioplasty which is shown by ChIPseq analysis. Figure 1A convincingly shows an increase in the given histone mark at certain TSS (cluster 1). However, the text of the related figure is misleading and the authors should take the following into account:

2. It is mentioned several times in the text that H3K27ac and BRD4 are primarily enhancer marks. Yet, (i) Figure 1A focuses on TSS +/-5kb and (ii) both, the localization of H3K27ac and BRD4 has not only been shown on enhancers but also on TSS/promoter (e.g. Zhang et al. J Biol Chem 2012; Anders et al. Nat Biotech 2013) which is basically also verified in Figure 1A. Please carefully revise your conclusions. How can "enhancer-associated BRD4 peaks" be located at promoters?

>> We have carefully revised the description involving enhancers. Please see Page 4, Paragraph 2.

“To survey gene-activating chromatin remodeling, we performed ChIPseq using H3K27ac and BRD4 as chromatin marks that are associated with active enhancers....”

3. The genomic features associated with H3K27ac and H3K27me3 are nicely introduced but is missing for H3K4me1.

>> Thanks for the reminder. We now add description for H3K4me1, please see Page 4, the middle of Paragraph 3:

"Both BRD4 and H3K27ac peaks overlapped with that of H3K4me1. The total number of H3K4me1 peaks was greater, which is reasonable as H3K4me1 enriches not only at active enhancers but also inactive and poised enhancers."

4. Figure 1B - legend: There is no transcript abundance of genes shown here. Please revise.

>> Revised. Please see now Figure 2A legend:

"A. Bean plot showing genomewide distribution of BRD4 or histone mark ChIPseq peak values."

Moreover, in the text it says that "gene annotation showed that gene-regulatory factors are top scored in both (..) ". Please include the mentioned gene annotation in the presented figure and give a list of those factors. In the same line the analysis would benefit to a great extent from a correlation of the ChIPseq data to gene expression data (own data or from public database).

>> This is a great suggestion. We now present in Figure 1B the GO analysis result based on gene annotation, as also suggested by Reviewer#1. Please see revision in Page 4, the bottom of Paragraph 2:

"Gene annotation reveals that gene-regulatory factors are top scored in both Cluster-1 and Cluster-2 (Figure 1B, Figure S2)."

5. In general, it would improve the manuscript to elaborate on the ChIPseq data in more detail:

- not only focusing on promoter but also enhancer regions - are they regulated differently upon injury?

>> Please see revision in Page 5, the lower part of Paragraph 1:

"Moreover, injury-induced H3K27ac/BRD4 co-enrichment occurred at enhancers not only in intronic regions (Figure 3, Figure S4) but also in the upstream of TSS far from a promoter (see *Ccnd1*, Figure 3B)."

- including info about the genes next to certain epigenetic marks - are certain pathways affected by epigenetic remodeling? - motif enrichment in regions of interest?

>> Consistent with the GO pathway analysis (Figure 1B), EZH2 and UHRF1, both gene regulatory factors, were affected by epigenetic remodeling. Please see related new data in Figure 6, Figure 9J, and Figure 10. We analyzed but did not find injury-induced motif enrichment near these two genes.

The 3rd section focuses on the regulation of Ezh2 expression on the epigenetic level upon angioplasty. While the injury-induced enrichment of H3K27ac on the

Ezh2 promoter is clearly visible in the genome browser view (Figure 2A), it is less obvious for BRD4.

>> Thank you for pointing this out. We have improved data presentation with peak coverage values labeled on the IGV profile. Please see Figure 4B, and revision in Page 5, Paragraph 2:

“To this end, IGV profiles of ChIPseq data (Figure 4, A and B) provided an interesting clue of greater BRD4/H3K27ac occupancy (injured vs uninjured) at enhancers in *Ezh2* intronic and upstream regions.”

6. To support the corresponding statement in the text the authors are encouraged to include the quantification of the peak intensities.

>> Yes, thanks. We now add Table S1 to include peak intensities.

7. Please add qPCR data showing the induction of Ezh2 and Nrp2 upon injury.

>> qPCR data showing the induction of Ezh2 and Nrp2 upon injury are now presented in Figure S6A and Figure S1. Please see revision in Page 4, Paragraph 1:

“In keeping with the literature validating this IH-inducing model, NRP2 and UHRF1, two recently reported novel IH-prompting factors, were both upregulated in injured versus uninjured arteries at post-angioplasty day 7 (Figure S1).”

The authors claim that the peak region depicted in panel A (enriched for H3K27ac and BRD4) marks an active enhancer. However, the H3K4me1 ChIPseq data does not support that. The histone modification H3K4me1 is a common identifier of enhancers regions, independent of activity status.

>> The reviewer is correct. This point is now emphasized in the revision in Page 4, Paragraph 3:

“The total number of H3K4me1 peaks was greater, which is reasonable as H3K4me1 enriches not only at active enhancers but also inactive and poised enhancers”.

8. Adding ChIP data of H3K4me3 (promoter mark) would clarify the status of this region of interest.

>> Accomplished. We performed ChIP-qPCR new experiments, please see Figure 9J and description of the result in Page 7, Paragraph 3 (middle):

“Importantly, guided by the in vivo ChIPseq data (Figures 3 and 4), the experiments of ChIP-qPCR using the in vitro SMC model demonstrated mitogen-induced H3K27ac enrichment at *Ezh2* and *Uhrf1* (Figure 9J), bridging in vitro and in vivo observations.”

9. In order to exclude that the used sgRNA targeting the region of interest is showing an effect independent of interfering with transcription, the reviewer recommends to add data of a sgRNA targeting an intronic region further

upstream as a control.

>> This is a nice suggestion about negative control, thanks. Please see new Figure S5, and revision in Page 5, middle of Paragraph 2:

“In a negative control experiment, EZH2 mRNA was not reduced when we applied sgRNAs that targeted an upstream region very much away (~50 kb) from the *Ezh2* TSS (Figure S5).”

The 4th section shows that SMC-specific deletion of Brd4 reduces IH and results in decreased H3K27me3 upon injury.

10. The deletion efficiency by TAM administration needs to be quantified.

>> Accomplished. Tamoxifen-induced BRD4 deletion is now shown in new Figure 5C.

11. In the same line, to support the link to *Ezh2*, please add the expression of *Ezh2* upon injury in general and upon Brd4 deletion in SMC in vivo.

>> Accomplished. For the expression of *Ezh2* upon injury in general, please refer to Figure S6. As for its expression upon Brd4 deletion in SMC in vivo, please see Figure 6A/B and revision in Page 6, Paragraph 2:

“We then measured on artery cross-sections the levels of EZH2 (Figure 6, A and B) and its catalytic product H3K27me3 (Figure 6, C and D). Immunofluorescence indicated that BRD4 knockout substantially reduced EZH2 in the neointimal layer;”

12. The last part of this results section refers to in vitro data that is currently presented in Figure 4. In order to keep the structure consistent, these data would be more suitable in Figure 3.

>> Fixed. The in vitro figures are now put together in new Figure 4.

The following 5th section addresses the role of *Ezh2* in IH formation by gain and loss of function experiments convincingly showing that *Ezh2* promotes IH.

13. In order to verify that the effects shown in panel G-I are *Ezh*-dependent, the authors should verify the expression of *Ezh1* and *Ezh2* either by (GFP) staining or expression quantification of isolated SMCs from arteries.

>> We agree. The same lenti-vectors resulted in expressed GFP-tagged EZH1 and EZH2 proteins in vitro in SMCs (Figure 8). We tried two GFP antibodies in vivo but none produced good specific immunofluorescent signal. We therefore decided to immunostain the EZH1/2 catalytic product H3K27me3. Please see new Figure 7B and revision in Page 6, Paragraph 3 (middle):

“We found that compared to the GFP control, increasing EZH2 heightened H3K27me3 (Figure 7B) and exacerbated IH and restenosis (lumen narrowing) (Figure 7C).”

14. In the legend: please explain the abbreviations N and M used in panel E.

>> Done. Thanks.

In the 6th section the manuscript focuses on in vitro studies to underline the role of both Ezh1 and Ezh2 in SMC proliferation and migration induced by PDGFB. This reviewer encourages the authors to address the following points:
15. The picture in panel B depicting the 0h time point should be replaced by a more convincing picture (GFP staining)

>> We used calcein to render cells green fluorescent at the end of experiment (e.g. 24h). We did not add calcein at the 0h time point (no color) and only took bright-field pictures. This is now clarified in the Figure S8 legend.

16. Panel C and F: Comparing the control samples (scrambled and Lenti-GFP) the treatment with PDGFB induces K27me3 in panel C while it not induced (rather repressed) in panel F. Please comment on that.

>> Thanks for the reminder. We now include a representative blot, please see Figure 8E.

17. In the same line, the control treatments in Panel E and H should be same/similar in migration/ scratch closure - but they are remarkable different. Please comment on that as well.

>> Thanks. Please see better representative pictures in Figure 8, C and G.

Section #7 (Upper Part of Figure 6) claims Uhrf1 as a novel target of Ezh isoforms. This finding is linked to the first figure and should rather be presented as a supplement of it. Moreover, the link has been published previously.

>> Thanks to your suggestion, we now present *Uhrf1* gene tracks in Figure 3, for a better ChIPseq data context.

Although literature search indicated that EZH2 and UHRF1 have been previously linked, whether EZH2 regulates UHRF1 expression was not determined. Please see revision to stress this point, in Page 9, Paragraph 2:

“UHRF1 and EZH2 were previously linked for their paralleled functions in keratinocyte self-renewal and for their positive correlation in human prostate tumor samples. However, whether EZH2 and/or BRD4 impose an epigenetic control over UHRF1 was not known.”

The last section (Fig. 6 E-G and Fig. 7) addresses the redistribution of H3K27me3 upon angioplasty. A point that has been raised in the first figure. Therefore, the authors should consider to shift these analyses to the first part of the manuscript and demonstrate the re-distribution on a genomic scale of H3K27ac/me3 rather on single gene expression level.

>> This is an excellent suggestion. We now put together the relevant gene tracks in Figure 3, and accordingly move the corresponding contents to the earlier part of the manuscript. Please see Page 5, Paragraph 1:

“As illustrated by Figure 3A and 3B (ChIPseq peak coverage presented in Table S1), H3K27me3 ChIPseq peaks intensified at *P57* but ebbed at *Ccnd1* after arterial injury; in contrast, H3K27ac occupancy markedly increased instead at *Ccnd1*. Moreover, injury-induced H3K27me3-up/H3K27ac-down were found at the gene of BMP4 (Figure S4), an anti-proliferative factor in SMCs that counters IH, and H3K27ac-up/H3K27me3-down occurred at the gene loci of other pro-proliferative factors including UHRF1 (Figure 3C) and NRP2 (Figure S4), both recently reported to promote SMC and neointima proliferation.”

Minor comments:

Figure legends:

- technical details in legends should be removed and described in the methods section only

>> Done. For example, some details are moved to Page 13, Paragraph 2:

“Cells were starved with 0.5% FBS overnight and then stimulated with PDGF-BB (20 ng/ml). At 72 h of PDGF-BB treatment, plates were decanted.”

- partially missing n numbers (qPCR data, WB data)

>> Added to figure legends.

Results section:

- The concluding remarks after each results section is partially missing.

>> Now added. Thanks.

- Some sentences set the results in perspective with other data which is rather discussion

>> Please see some sentences moved to Discussion, Page 9, Paragraph 2:

“Epigenetic players have become increasingly appreciated for their importance in vascular homeostasis and dysregulation, yet their functional relationships remain poorly interpreted. To this end, herein we identified angioplasty-induced genomewide chromatin remodeling that entails a BRD4->EZH2->UHRF1 regulatory cascade.”

Methods:

Please deposit your ChIPseq data in public database (like GEO) to make it available and a valid source for further research.

>> It is being done according to the journal instructions.

At least the peak regions should be provided as a supplement (e.g bed file format).

>> Please find Table S7, the Excel file.

>> Thank you very much for helping us to improve the science and the manuscript.

January 4, 2022

RE: Life Science Alliance Manuscript #LSA-2021-01114R

Dr. Lian-Wang Guo
University of Virginia
409 Lane Rd, MR4(building), Room 2146
Charlottesville, VA 22908

Dear Dr. Guo,

Thank you for submitting your revised manuscript entitled "Angioplasty induces epigenomic remodeling in injured arteries". We would be happy to publish your paper in Life Science Alliance pending final revisions necessary to meet our formatting guidelines.

- According to Reviewer 3 comments, please edit the wording in the manuscript concerning the usage of the term "enhancer" as indicated in section 2, page 4 and in the text referring to the new figure 4A
- Please add Artery tissue ChIP sequencing accession number to the Data availability section
- please add ORCID ID for secondary corresponding author-they should have received instructions on how to do so
- please use the [10 author names, et al.] format in your references (i.e. limit the author names to the first 10)
- please add your main, supplementary figure, and table legends to the main manuscript text after the references section
- all figure legends should only appear in the main manuscript file
- Please indicate molecular weight next to each protein blot
- please upload your Tables in editable .doc or excel format
- please add callouts for Figures 4G, S3A-C, S6A-B, S8A-B to your main manuscript text

FIGURE CHECKS:

- scale bars for figures 5C and D; 6A and C; 10A and C are missing, please provide them
- the Source Data provided for Figure 8E (EZH1 in figure 8E) do not match. Please provide the correct source data for this blot

A. FINAL FILES:

B. MANUSCRIPT ORGANIZATION AND FORMATTING:

Sincerely,

Reviewer #1 (Comments to the Authors (Required)):

The authors have sufficiently addressed my concerns.

Reviewer #2 (Comments to the Authors (Required)):

The authors performed an excellent revision

Reviewer #3 (Comments to the Authors (Required)):

The authors have substantially edited the manuscript. Clearly, rephrasing and restructuring has further stressed the central theme and the novel findings. Furthermore, the edited presentation of already existing data and the inclusion of additional data have all contributed to strongly improve the manuscript.

The authors have addressed most of my comments, but missed a critical point I made.

Based on the following comments the authors are encouraged to carefully edit the wording in the manuscript.

Comments:

1. Based on the 2nd comment of this reviewer, the authors have tried to revise the manuscript concerning the usage of the term "enhancer". Yet, the authors miss to clarify that some of the regions they focus on might be promoter and not enhancer regions. The most well-established histone marks to distinguish promoter from enhancer regions are H3K4me3 and H3K4me1, respectively. Still, it is well accepted that H3K4me1 is found at both active promoters and enhancers: while H3K4me3 localizes closest to the TSS, H3K4me1 spreads furthest downstream (e.g. Barski et al. 2007 Cell). In line with that notion, the plots shown in Figure 1A show that BRD4 and H3K27ac enrichment are highest on the TSS (the center) and the enhancer mark H3K4me1 is rather flanking the TSS.

As the manuscript contains no H3K4me3 data, promoter and enhancer regions cannot be perfectly separated. This should be made clear in the manuscript. For instance, to add "...and promoters" to their following comment would avoid mis-interpretations

(Page 4, section2):

"To survey gene-activating chromatin remodeling, we performed ChIPseq using H3K27ac and BRD4 as chromatin marks that are associated with active enhancers...."

2. In the same line, this reviewer had the following comment which the authors did not refer to (referring to the new Figure 4A): The authors claim that the peak region depicted in panel A (enriched for H3K27ac and BRD4) marks an active enhancer. However, the H3K4me1 ChIPseq data does not support that. As there is no H3K4me1 signal visible at the indicated yellow boxes (which might also be a scaling issue?), the authors cannot claim these as enhancer regions.

Dear Dr. Guidi,

We would like to thank you for considering our manuscript for publication. We appreciate very much the efforts and valuable comments from the reviewers that greatly helped improve the manuscript. Please find below our responses to address the remaining points. We hope that the manuscript is now acceptable for publication.

Thank you again.

Lian-Wang Guo, PhD, University of Virginia

From: "n.guidi@life-science-alliance.org" <n.guidi@life-science-alliance.org>
Reply-To: "n.guidi@life-science-alliance.org" <n.guidi@life-science-alliance.org>
Date: Tuesday, January 4, 2022 at 5:43 AM
To: "Guo, Lian-Wang (lg8zr)" <lg8zr@virginia.edu>
Subject: Life Science Alliance Manuscript - Editorial Decision LSA-2021-01114R

January 4, 2022

RE: Life Science Alliance Manuscript #LSA-2021-01114R

Dr. Lian-Wang Guo
University of Virginia
409 Lane Rd, MR4(building), Room 2146
Charlottesville, VA 22908

Dear Dr. Guo,

Thank you for submitting your revised manuscript entitled "Angioplasty induces epigenomic remodeling in injured arteries". We would be happy to publish your paper in Life Science Alliance pending final revisions necessary to meet our formatting guidelines.

- According to Reviewer 3 comments, please edit the wording in the manuscript concerning the usage of the term "enhancer" as indicated in section 2, page 4 and in the text referring to the new figure 4A

>> We agree with the reviewer and have made revisions accordingly.
For our response to Reviewer#3's first comment, please see page 4 red letters (now section 3).
For the response to Reviewer#3's second comment, we revised the Figure 4 legend.

- Please add Artery tissue ChIP sequencing accession number to the Data availability section
>> Added.

-please add ORCID ID for secondary corresponding author-they should have received instructions on how to do so.

>> Done. Thanks.

-please use the [10 author names, et al.] format in your references (i.e. limit the author names to the first 10)

>> All cited references have been changed to the LSA journal style.

-please add your main, supplementary figure, and table legends to the main manuscript text after the references section

>> Done.

-all figure legends should only appear in the main manuscript file

>> Done.

-Please indicate molecular weight next to each protein blot

>> Yes, added.

-please upload your Tables in editable .doc or excel format

>> Uploaded.

-please add callouts for Figures 4G, S3A-C, S6A-B, S8A-B to your main manuscript text

>> Added. Please see red letters.

FIGURE CHECKS:

-scale bars for figures 5C and D; 6A and C; 10A and C are missing, please provide them

>> Provided.

-the Source Data provided for Figure 8E (EZH1 in figure 8E) do not match. Please provide the correct source data for this blot

>> The provided original Western blot is the right source data. It doesn't appear to match because we are not including the upper band in the main figure (EZH1 in figure 8E), for the consideration that we are not sure whether the upper band is an oligomer of the endogenous EZH1 protein or merely non-specific signal.

>> N/A.

To upload the final version of your manuscript, please log in to your account: <https://lsa.msubmit.net/cgi-bin/main.plex>

A. FINAL FILES:

B. MANUSCRIPT ORGANIZATION AND FORMATTING:

Sincerely,

Reviewer #1 (Comments to the Authors (Required)):

The authors have sufficiently addressed my concerns.

Reviewer #2 (Comments to the Authors (Required)):

The authors performed an excellent revision

Reviewer #3 (Comments to the Authors (Required)):

The authors have substantially edited the manuscript. Clearly, rephrasing and restructuring has further stressed the central theme and the novel findings. Furthermore, the edited presentation of already existing data and the inclusion of additional data have all contributed to strongly improve the manuscript.

The authors have addressed most of my comments, but missed a critical point I made. Based on the following comments the authors are encouraged to carefully edit the wording in the manuscript.

Comments:

1. Based on the 2nd comment of this reviewer, the authors have tried to revise the manuscript concerning the usage of the term "enhancer". Yet, the authors miss to clarify that some of the regions they focus on might be promoter and not enhancer regions. The most well-established histone marks to distinguish promoter from enhancer regions are H3K4me3 and H3K4me1, respectively. Still, it is well accepted that H3K4me1 is found at both active promoters and enhancers: while H3K4me3 localizes closest to the TSS, H3K4me1 spreads furthest downstream (e.g. Barski et al. 2007 Cell). In line with that notion, the plots shown in Figure 1A show that BRD4 and H3K27ac enrichment are highest on the TSS (the center) and the enhancer mark H3K4me1 is rather flanking the TSS.

As the manuscript contains no H3K4me3 data, promoter and enhancer regions cannot be perfectly separated. This should be made clear in the manuscript. For instance, to add "...and promoters" to their following comment would avoid mis-interpretations (Page 4, section2): "To survey gene-activating chromatin remodeling, we performed ChIPseq using H3K27ac and BRD4 as chromatin marks that are associated with active enhancers..."

2. In the same line, this reviewer had the following comment which the authors did not refer to (referring to the new Figure 4A):

The authors claim that the peak region depicted in panel A (enriched for H3K27ac and BRD4) marks an active enhancer. However, the H3K4me1 ChIPseq data does not support that. As there is no H3K4me1 signal visible at the indicated yellow boxes (which might also be a scaling issue?), the authors cannot claim these as enhancer regions.

January 26, 2022

RE: Life Science Alliance Manuscript #LSA-2021-01114RR

Dr. Lian-Wang Guo
University of Virginia
409 Lane Rd, MR4(building), Room 2146
Charlottesville, VA 22908

Dear Dr. Guo,

Thank you for submitting your Research Article entitled "Angioplasty induces epigenomic remodeling in injured arteries". It is a pleasure to let you know that your manuscript is now accepted for publication in Life Science Alliance. Congratulations on this interesting work.

DISTRIBUTION OF MATERIALS:

Again, congratulations on a very nice paper. I hope you found the review process to be constructive and are pleased with how the manuscript was handled editorially. We look forward to future exciting submissions from your lab.

Sincerely,
